# Direct biological fixation provides a freshwater sink for $N_2O$

Yueyue Si [1], Yizhu Zhu [1], Ian Sanders[1], Dorothee B. Kinkel[1], Kevin J. Purdy [2] & Mark Trimmer [1] ✉

Nitrous oxide ($N_2O$) is a potent climate gas, with its strong warming potential and ozone-depleting properties both focusing research on $N_2O$ sources. Although a sink for $N_2O$ through biological fixation has been observed in the Pacific, the regulation of $N_2O$-fixation compared to canonical $N_2$-fixation is unknown. Here we show that both $N_2O$ and $N_2$ can be fixed by freshwater communities but with distinct seasonalities and temperature dependencies. $N_2O$ fixation appears less sensitive to temperature than $N_2$ fixation, driving a strong sink for $N_2O$ in colder months. Moreover, by quantifying both $N_2O$ and $N_2$ fixation we show that, rather than $N_2O$ being first reduced to $N_2$ through denitrification, $N_2O$ fixation is direct and could explain the widely reported $N_2O$ sinks in natural waters. Analysis of the nitrogenase (*nifH*) community suggests that while only a subset is potentially capable of fixing $N_2O$ they maintain a strong, freshwater sink for $N_2O$ that could be eroded by warming.

Nitrous oxide ($N_2O$) is a potent climate gas, with ~273 times the global warming potential of carbon dioxide ($CO_2$)[1] and strong ozone-depleting properties[2]. The atmospheric concentration of $N_2O$ continues to rise through the use of nitrogen-based fertilisers, fossil fuel combustion, biomass burning and sewage discharge[3] and has already increased by approximately 20% since 1750[4]. Not surprisingly, given its atmospheric potency, research to date has focused on these $N_2O$ sources with $N_2O$ sinks being relatively understudied[5–7]. The few studies reporting on both $N_2O$ sources and sinks[8–11] often simply document the sinks as concentrations below that expected for water (marine or freshwater) at equilibrium with the atmosphere and the true mechanism behind this $N_2O$ deficit remains largely unknown.

In terrestrial and aquatic environments, $N_2O$ can be produced from both microbial nitrification[12] either via hydroxylamine oxidation ($NH_4^+ \rightarrow NH_2OH \rightarrow N_2O$), or hybrid formation ($NO_2^- + NH_2OH \rightarrow N_2O$)[13], and incomplete denitrification ($NO_3^- \rightarrow NO_2^- \rightarrow NO \rightarrow N_2O[\rightarrow N_2]$)[14]. Where oxygen is limiting and/or completely absent, $N_2O$ can be further reduced to $N_2$ in the last step of microbial denitrification ($N_2O \rightarrow N_2$) that is typically mediated by facultative anaerobic bacteria[14,15]. As such, any undersaturation – indicating a sink for $N_2O$ – as observed in some waters has routinely been attributed to that last step in denitrification.

However, such $N_2O$ undersaturation has typically been reported in well-oxygenated, shallow freshwaters[8–10,16–21] (down to 13% of air equilibration, typically ~70–100%) and surface-ocean-waters[5,11,22–28] (down to 34%, typically ~90%) where canonical denitrification is unlikely to explain any undersaturation in $N_2O$. While $N_2O$ consumption by denitrification has been reported in both anoxic and oxic-to-anoxic transitioning waters in the Eastern Tropical North Pacific[6], the reasons for $N_2O$ undersaturation in general remain poorly understood, with many instances of $N_2O$ undersaturation remaining unaccounted for[8–10,19–21] or simply being dismissed as analytical artifacts[24,29]. Further, as $N_2O$ sources generally increase at higher concentrations of ammonium and nitrate (i.e., fixed, bio-available N)[8,25], any potential undersaturation in $N_2O$ could be masked by stronger production of $N_2O$ from nitrification and denitrification. This might explain why many accounts of $N_2O$ undersaturation have been reported in N limited environments[5,9,19,21,22].

In recent years, evidence has been presented for an additional pathway to denitrification for $N_2O$ reduction, namely – $N_2O$ dependent N fixation – that has been reported for pure cultures of marine *Trichodesmium* and *Crocosphaera*[5]. $N_2O$ fixation has also been reported in the surface waters of the Eastern Tropical South Pacific[5,23], where the

[1]School of Biological and Behavioural Sciences, Queen Mary, University of London, London E1 4NS, UK. [2]School of Life Sciences, University of Warwick, Coventry CV4 7AL, UK. ✉e-mail: m.trimmer@qmul.ac.uk

measured N$_2$O fixation activity could contribute some (0.2 – 60%) of the total N$_2$O reduction[5]. As long ago as 1954, it was shown[30] that $^{15}$N$_2$O could be assimilated by soybean root nodules with activity comparable to $^{15}$N$_2$ assimilation. These findings show that N$_2$O fixation (e.g. N$_2$O→NH$_4^+$) represents an alternative N$_2$O reduction pathway to the terminal step in denitrification (N$_2$O→N$_2$) that may explain some of the undersaturation reported for N$_2$O. Within the widespread accounts of N$_2$O undersaturation found in well-oxygenated waters, only a few studies mentioned the possibility of N$_2$O fixation[5,22,23] and it is not widely acknowledged.

Primary production and N$_2$ fixation are tightly coupled in N-limited ecosystems[31]. Some early studies (1952–1986) showed that N$_2$O is a competitive inhibitor for N$_2$ fixation and it could also be a substrate for the enzymatic nitrogenase complex[32–35], indicating that N$_2$O fixation (e.g. N$_2$O→NH$_4^+$) may be related to N$_2$ fixation (N$_2$→NH$_4^+$). Further, with its N≡N bond N$_2$ fixation has a high activation energy (~1 to 2 eV vs. 0.65 eV and 0.32 eV for respiration and photosynthesis, respectively)[36,37] which makes fixing N$_2$ in the cold energetically unfavourable. As a consequence, the abundance of diazotrophs has been shown to decrease as temperatures decline[36]. In contrast, the energy required to fix N$_2$O (Eq. 1, $\Delta G$ defined for freshwater at 10 °C, see Supplementary Text 1) is lower than that for N$_2$ (Eq. 2) and being able to fix N$_2$O could confer an ecological advantage to some microbes either in the cold or when resources (light or reduced substrates) in general are limiting.

$$0.5N_2O + 1.5H_2O \rightarrow 1NH_3 + 1O_2 \quad \Delta G = +247kJ \quad (per\ NH_3) \quad (1)$$

$$0.5N_2 + 1.5H_2O \rightarrow 1NH_3 + 0.75O_2 \quad \Delta G = +291kJ \quad (per\ NH_3) \quad (2)$$

While the ~18% energy saving for fixing N$_2$O versus N$_2$ is seemingly modest, it is comparable to the recognised 21% saving delivered by assimilating NO$_3^-$ (Eq. 3) rather than fixing N$_2$ (ref. 38) (see Supplementary Text 1).

$$NO_3^- + 3H^+ + 2e^- \rightarrow 1NH_3 + 1.5O_2 \quad \Delta G = +241kJ \quad (per\ NH_3) \quad (3)$$

With both the last step in denitrification (N$_2$O→N$_2$) and N$_2$O fixation (N$_2$O→NH$_4^+$) providing sinks for N$_2$O it is ecologically important to distinguish between these two parts of total N$_2$O reduction. Further, any genuine direct N$_2$O fixation (N$_2$O→NH$_4^+$) needs to be distinguished from indirect N$_2$O fixation i.e., that which could occur after the initial reduction of N$_2$O to N$_2$ (N$_2$O→N$_2$→NH$_4^+$). Despite the few studies[5,23,30] documenting N$_2$O fixation so far, to the best of our knowledge, there has been no characterisation of N$_2$O fixation in relation to canonical N$_2$ fixation through the dual use of $^{15}$N$_2$O and $^{15}$N$_2$ in natural communities.

In 2005, we set up 20 experimental ponds (each with 1 m$^3$ water volume, 0.5 m depth) in East Stoke, Dorset, UK, to experimentally study the whole-ecosystem effects of climate warming[39–41]. Here, however, we exploited the fact that our experimental ponds are also N-limited[41], being fed only by rain water, to characterise any potential N$_2$O fixation in a controlled, experimental system. Despite being artificial, the ponds have well-established freshwater ecosystems[39–42] with diverse cyanobacteria communities[42], among which some Nostocales[43] and Oscillatoriales[44] are known to fix N$_2$.

Here, we show that the ponds are undersaturated in both N$_2$ and N$_2$O and further hypothesise that the pond communities fix both gases to support primary production. Then, due to the different energy demands of N$_2$ and N$_2$O fixation, we hypothesise that the two processes will respond differently to temperature. We use incubations with pond biomass and $^{15}$N$_2$ and $^{15}$N$_2$O stable isotope techniques to quantify their fixation activity, distinguish direct from indirect N$_2$O fixation and characterise the temperature dependence of each N-fixing process. Finally, with no known freshwater candidates for N$_2$O fixation

to date, we explore the recognised N$_2$ fixing community in relation to N$_2$O fixation. We ask whether: 1, is N$_2$O fixation mediated by the total nitrogenase (nifH) community simply in relation to the relative availability of N$_2$O to N$_2$; or 2, is N$_2$O fixation preferentially mediated by a subset of the nifH community?

## Results

### Contrasting seasonalities in undersaturation for N$_2$ and N$_2$O

Concentrations of dissolved N$_2$O and N$_2$ were both significantly below atmospheric equilibrium ($p < 0.001$, Fig. 1a) and the ponds are sinks for both atmospheric N$_2$O and N$_2$. Overall, N$_2$O was more undersaturated than N$_2$ ($p < 0.001$, $t = -17.5$, d.f. = 240.6, two-sided, Fig. 1a), with a mean value of 79.1% ± 1.1% (mean ± s.e., as below) of air saturation compared to 98.5% ± 0.2% for N$_2$. Furthermore, the seasonality in N$_2$O saturation was far more pronounced than for N$_2$ (Best fitting Generalised Additive Mixed Models, GAMMs, Supplementary Table 1), with a strong minimum for N$_2$O in December and maximum saturation in summer (Fig. 1b). Conversely, N$_2$ saturation peaked in winter and was lower in spring and summer (Fig. 1c).

The concentrations of dissolved inorganic nutrients (nitrite, NO$_2^-$; nitrate, NO$_3^-$; ammonium, NH$_4^+$, and soluble reactive phosphorus, SRP) were low in the ponds, with NO$_2^-$, NO$_3^-$ and NH$_4^+$ often at or below the limit of detection. The concentration of total inorganic nitrogen (TIN as the sum of NO$_2^-$, NO$_3^-$ and NH$_4^+$) was 0.85 ± 0.03 μM across all sampling months (Fig. 1f). SRP concentrations were 0.14 ± 0.01 μM, on average, and, at 5 to 1, the median N to P ratio was markedly lower than Redfield[45] (16 to 1), indicating primary production in the ponds to be N limited (Fig. 1g). As the ponds were N-limited, primary production must be sustained largely by N fixation (and any unknown atmospheric N deposition), which may have resulted in the undersaturation of N$_2$ and N$_2$O in the ponds.

Interestingly, N$_2$O saturation increased with water temperature ($p < 0.001$, Fig. 1d), suggesting relatively higher net reduction of N$_2$O in the cold (see Supplementary Fig. 1 for concentration data). Whereas N$_2$ saturation showed the opposite pattern, with relatively more net N$_2$ reduction at higher temperatures ($p < 0.001$, Fig. 1e) in spring and summer. Moreover, the saturation of dissolved O$_2$ in the ponds (at the same depth where the samples for N$_2$ and N$_2$O were collected) was generally around air-equilibration (104.8% ± 1.8%, median 99.6%), with N$_2$ saturation decreasing at higher O$_2$ saturations, while N$_2$O saturation increased with higher O$_2$ saturation (Supplementary Fig. 2). Oxygen saturation was positively correlated with temperature (Supplementary Fig. 2c), probably due to higher temperatures in spring and summer promoting primary production. Therefore, maximum N$_2$ undersaturation was probably related to higher primary production in spring and summer[40]. The negative and positive correlations between N$_2$ or N$_2$O and O$_2$ respectively, indicated different controls for the reduction of N$_2$ and N$_2$O.

### N$_2$O and N$_2$ fixation by biomass in the ponds

To rationalise the undersaturation in both N$_2$O and N$_2$ in our ponds, we measured fixation of either $^{15}$N$_2$O or $^{15}$N$_2$ (at a range of temperatures, see below) by biomass collected from the ponds (Supplementary Fig. 3). We found $^{15}$N assimilated into biomass from either $^{15}$N$_2$ or $^{15}$N$_2$O in the majority of our incubations (87%, 572 out of 658 incubations, Fig. 2a), with higher rates of $^{15}$N assimilation with $^{15}$N$_2$ than for $^{15}$N$_2$O with both floating and benthic biomass ($p < 0.001$, $t = 6.5$, d.f. = 369.7, two-sided, Fig. 2b). On average, 11.5 ± 0.9 and 5.3 ± 0.3 nmol g$^{-1}$ d$^{-1}$ (mean ± s.e.) of $^{15}$N were assimilated into biomass with either $^{15}$N$_2$ or $^{15}$N$_2$O, respectively (Fig. 2b). The rate of $^{15}$N$_2$ assimilation was higher in the floating than the benthic biomass ($p = 0.001$, $t = 3.3$, d.f. = 179.4, two-sided), while $^{15}$N$_2$O assimilation was consistent between the two biomass types ($p = 0.24$, $t = -1.2$, d.f. = 319.1, two-sided).

To distinguish direct N$_2$O fixation (N$_2$O → NH$_4^+$, Eq. 1) from indirect fixation i.e., that after an initial reduction of N$_2$O to N$_2$ ([N$_2$O → N$_2$]

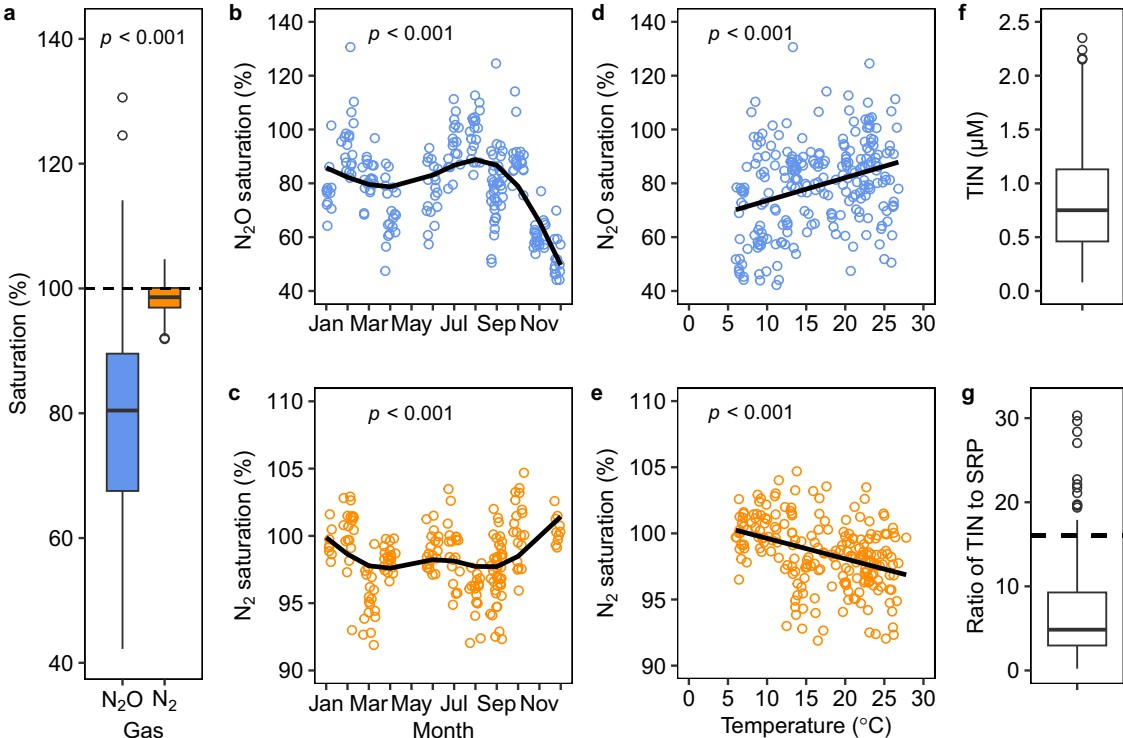

**Fig. 1 | Seasonal and overall levels of N₂O and N₂ saturation in the ponds. a** Box-whisker plots showing overall that the saturation of N₂O was lower than that for N₂ in the ponds ($p < 0.001$, two-sided, Supplementary Table 1). The dashed line denotes 100% atmospheric equilibrium for the gases. **b, c** show the saturation in N₂O had a different seasonal pattern compared to N₂ (Note the different scales on the y-axes). The solid lines in **b** and **c** represent the best fitting GAMM models (two-sided, Supplementary Table 1). **d** N₂O saturation increased at higher temperatures while, in contrast, **e** N₂ saturation declined. The lines in **d** and **e** are simple first-order linear regressions (two-sided). **f** Overall concentration of total inorganic nitrogen (TIN) in the ponds ($n = 213$ samples for 11 months in 20 ponds). **g** The ratio of TIN to soluble reactive phosphorous (SRP) in the ponds ($n = 168$ samples for 11 months for 20 ponds, SRP data omitted below the detection limit). The dashed line in **g** denotes the Redfield ratio of N to P of 16:1. Each box in **a**, **f** and **g** shows the 25th to 75th percentiles, horizontal lines the median, open circles denote outliers and whiskers extend to 1.5 times the interquartile range. $n = 230$ and $n = 215$ samples for N₂O and N₂ saturation, respectively, for 20 ponds, in 11 months from November 2019 to April 2022 (see Supplementary Table 2).

N₂ → NH₄⁺, Eq. 2) through denitrification, we first checked for any production of ¹⁵N₂ from ¹⁵N₂O. Overall, the production of ¹⁵N₂ in the ¹⁵N₂O treatments was not significant in the floating biomass incubations (Fig. 2c), though ¹⁵N assimilation from ¹⁵N₂O was significant (Fig. 2b). In contrast, 30% of the benthic incubations showed measurable ¹⁵N₂ production ($p = 0.04$, two-sided, Fig. 2c and Supplementary Fig. 5) but with comparable ¹⁵N assimilation from ¹⁵N₂O (Fig. 2b). Some denitrification is expected given the sediments recognised capacity to consume oxygen[39], and our 12 h/12 h light/dark incubation-cycle generated oxygen minima overnight that likely facilitated the reduction of N₂O to N₂ via denitrification.

In addition, we also compared rates of assimilation against a theoretical upper threshold for indirect assimilation of ¹⁵N₂O after reduction to ¹⁵N₂ (Table 1 and Fig. 2b). For example, any ¹⁵N₂ from the reduction of ¹⁵N₂O would be assimilated in proportion to the ¹⁵N-labelling of the total N₂ pool, which would be predominantly ambient ¹⁴N₂ (Table 1). As a result, any indirect assimilation of ¹⁵N from ¹⁵N₂O should have been ~14-fold lower than what we measured in the incubations where we added ¹⁵N₂ directly e.g. 0.8 nmol N g⁻¹ d⁻¹ vs. 11.5 nmol N g⁻¹ d⁻¹ (Table 1). In contrast, we measured far higher rates of 5.3 nmol N g⁻¹ d⁻¹ with ¹⁵N₂O, compared to the upper threshold of 0.8 nmol N g⁻¹ d⁻¹, on average (0.69 to 0.92 nmol N g⁻¹ d⁻¹, 95% C.I., Fig. 2b). Such disproportionately high activity suggests direct assimilation of ¹⁵N from ¹⁵N₂O into biomass in our freshwater ponds.

Here we added ¹⁵N₂O to our incubations at concentrations many times higher than atmospheric equilibrium (9 μM vs. 0.01 μM) and our rates of ¹⁵N₂O assimilation are likely upper-potentials. We also characterised the kinetic effect of N₂O concentration on total N₂O

reduction from 9.2 nM (atmospheric equilibration) to 20,000 nM (Supplementary Fig. 6), which enabled us to estimate N₂O reduction by biomass at in situ concentrations in the ponds. We then scaled these in situ N₂O reduction estimates by the amount of benthic biomass in the ponds and compared them to our estimates of N₂O flux into the ponds calculated using our measurements of N₂O saturation (Fig. 1 and see Supplementary Text 2). Accordingly, we estimated in situ N₂O reduction by the benthic biomass to be −0.75 μmol N₂O m⁻² d⁻¹ (Supplementary Text 2) which is equivalent to 56% of the N₂O flux into the ponds of −1.33 μmol N₂O m⁻² d⁻¹, on average (range of −3.65 to 0.02 μmol N₂O m⁻² d⁻¹, including low emissions to the atmosphere in summer). The remaining ~44% of the N₂O flux is probably driven by microbes associated with the floating biomass (Fig. 2b) or free-living in the water column[42] and we are confident that our laboratory biomass incubations can rationalise the undersaturation in N₂O we measured in our ponds. In addition, N₂ flux into the ponds was −3,934 μmol N₂ m⁻² d⁻¹, on average (Supplementary Text 2).

## Multiple fates for total ¹⁵N₂O reduction
Apart from ¹⁵N₂O being assimilated into biomass and the fraction reduced to ¹⁵N₂ (above), some fixed ¹⁵N₂O as ¹⁵NH₄⁺ could potentially "leak" into the pond-water medium to, in turn, be nitrified to ¹⁵NO₂⁻ and ¹⁵NO₃⁻ (together ¹⁵NO_x⁻) – all of which comprise total ¹⁵N₂O reduction. We characterised total ¹⁵N₂O reduction and the proportions of the different end-products and found significant ¹⁵N₂O reduction in the majority (289 out of 372 incubations, 78%) of our incubations enriched with ¹⁵N₂O. The mean rate of total ¹⁵N₂O reduction was 364 ± 27 nmol N g⁻¹ d⁻¹, with the highest rate of total ¹⁵N₂O reduction occurring in

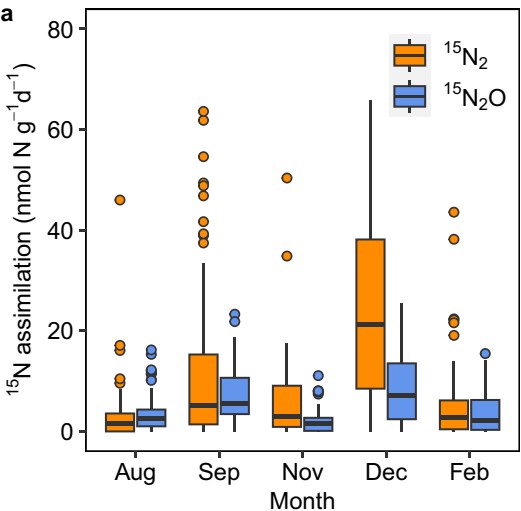

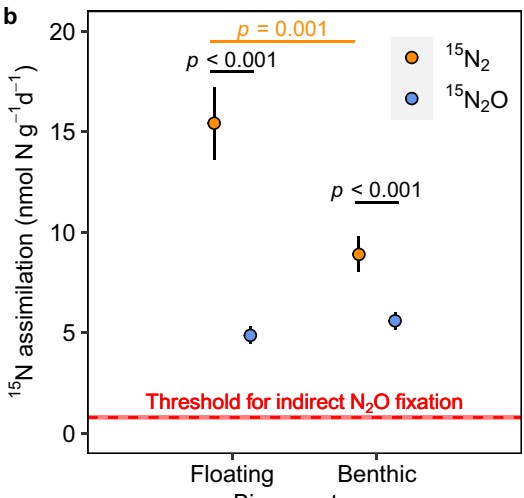

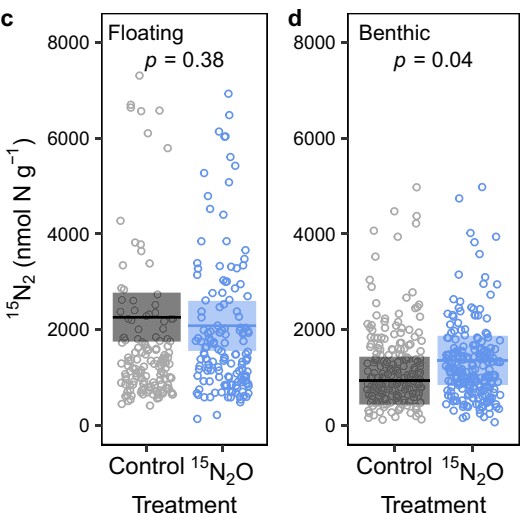

**Fig. 2 | Evidence for the direct assimilation of $^{15}$N into biomass from either $^{15}$N$_2$ or $^{15}$N$_2$O.** $^{15}$N assimilated into biomass from either $^{15}$N$_2$ or $^{15}$N$_2$O in different months. **a** Each box shows the 25th to 75th percentiles, horizontal lines the median, open circles denote outliers, and whiskers extend to 1.5 times the interquartile range. **b** Rates of $^{15}$N$_2$ assimilation were higher than for $^{15}$N$_2$O assimilation in both biomass types. The red dashed line marks the upper threshold for indirect $^{15}$N assimilation if $^{15}$N$_2$O was first reduced to $^{15}$N$_2$ prior to assimilation (see Table 1), red shaded area 95% C.I. Data plotted are means ± s.e. Yellow: difference between floating and benthic biomass with $^{15}$N$_2$ ($p = 0.001$, two-sided), black: difference between $^{15}$N$_2$ and $^{15}$N$_2$O in floating or benthic biomass ($p < 0.001$, two-sided). To exclude extreme outliers the data plotted in **a** and **b** are 95% of the dataset (2.5–97.5% percentiles). Data in **a** and **b** were collected in 5 calendar months for 2 biomass types from 8 to 10 ponds per sampling date ($n = 303$ and $n = 322$ incubations for $^{15}$N$_2$ and $^{15}$N$_2$O treatments, respectively). **c** No significant production of $^{15}$N$_2$ from $^{15}$N$_2$O in the floating biomass, but **d** significant production of $^{15}$N$_2$ in the benthic biomass ($p = 0.04$, two-sided). The horizontal line is the mean and shaded area 95% C.I. ($n = 351$ incubations for both controls and $^{15}$N$_2$O treatments).

$t = -5$, d.f. = 68.6, two-sided) and benthic ($p = 0.02$, $t = -2.3$, d.f. = 148.5, two-sided) biomass (Fig. 3a). The patterns in total $^{15}$N$_2$O reduction measured in the incubations agreed with the seasonal pattern of N$_2$O saturation in the ponds (Fig. 1): overall, N$_2$O was consumed in both seasons and the ponds were net sinks for N$_2$O, with higher N$_2$O reduction in winter, corresponding to greater undersaturation in N$_2$O in winter.

To test whether N$_2$O initially fixed intracellularly as NH$_4^+$ (Eq. 1) could leak into the water i.e., to be available to the wider ecosystem, we performed additional incubations with samples for nutrient measurements (without formaldehyde, *see* methods). Although the concentration of NH$_4^+$ was often below the limit of detection for the colorimetric assay (-0.2 μM), the stronger signal for NH$_4^+$ with N$_2$O ($p = 0.03$) indicated some N$_2$O fixed as NH$_4^+$ could "leak". The concentration of total inorganic nitrogen (TIN) was, on average, 0.32 μM higher in incubations enriched with N$_2$O than the controls ($p = 0.001$, Fig. 3b). We also characterised the production of $^{15}$NO$_x^-$ from $^{15}$N$_2$O in December, 2020 (winter), when rates of total $^{15}$N$_2$O reduction were highest. The rate of $^{15}$NO$_x^-$ production was 280 ± 46 nmol g$^{-1}$ d$^{-1}$ (mean ± s.e.), which accounted for 11.7% (median) of total $^{15}$N$_2$O reduction (Fig. 3c). Together, these results show that some N$_2$O fixed as NH$_4^+$ can be lost to the water and further oxidised to NO$_x^-$ through nitrification both of which could be assimilated into PON by the wider community (Fig. 3d).

## The temperature dependence of N$_2$O fixation
As seasonal changes in temperature drove contrasting patterns in N$_2$O and N$_2$ saturation, we characterised the effect of temperature on N$_2$O and N$_2$ reduction by incubating biomass from the ponds at temperatures from 6°C to 25°C. Assimilation of $^{15}$N from $^{15}$N$_2$ increased at higher temperatures ($p = 0.005$, $t = 2.8$, d.f. = 301, Fig. 4a), with an estimated Q$_{10}$ of 1.38. In contrast, assimilation of $^{15}$N from $^{15}$N$_2$O was consistent across all temperatures with no discernible temperature sensitivity. The large variance in Fig. 4a may in part be due to simply normalising the $^{15}$N assimilation data to a unit of dry biomass in each incubation, whereas the communities responsible for N$_2$ or N$_2$O assimilation could be heterogeneous in the biomass samples and across different months of the year. In addition, rates of $^{15}$NO$_x^-$ production from $^{15}$N$_2$O were also consistent across incubation temperatures (Fig. 4b), which, again, suggested that N$_2$O fixation is not sensitive to temperature (i.e., Fig. 4a, b).

## *nifH* communities in relation to N$_2$O reduction
The fact that here N$_2$O fixation appears less sensitive to temperature than N$_2$ fixation supported our hypothesis that fixing N$_2$O is less energy demanding than fixing N$_2$. Here we aimed to address our question of whether N$_2$O fixation is mediated by the whole N$_2$ fixing

December for both floating (850 ± 178 nmol N g$^{-1}$ d$^{-1}$) and benthic biomass (784 ± 158 nmol N g$^{-1}$ d$^{-1}$).

To compare summer to winter, we pooled data from November, February and December, for winter, and August and September for summer. Total $^{15}$N$_2$O reduction was highest in winter at 507 ± 49 nmol N g$^{-1}$ d$^{-1}$, compared to 237 ± 21 nmol N g$^{-1}$ d$^{-1}$ in summer, on average ($p < 0.001$, $t = -5.1$, d.f. = 213.4, two-sided) in both floating ($p < 0.001$,

**Table 1 | Rationalising N$_2$O assimilation as direct N$_2$O fixation**

| Treatment | Process | Frequency of $^{15}$N-labelling Direct $F_{N2}$ and $F_{N2O}$ or indirect $F_{N2}'$ | $^{15}$N assimilation (nmol N g$^{-1}$ d$^{-1}$) |
|---|---|---|---|
| $^{15}$N$_2$ | Direct N$_2$ fixation | $F_{N2} = 0.018 = [^{15}9\mu M/(^{15}9\ \mu M + {}^{14}487\ \mu M)]^{..}$ | 11.5 |
| $^{15}$N$_2$O | Direct N$_2$O fixation | $F_{N2O} = 0.98 = [^{15}9\ \mu M/(^{15}9\ \mu M + {}^{14}0.01\ \mu M)]^{..}$ | 5.3 |
| $^{15}$N$_2$O | *Indirect N$_2$O fixation | $F_{N2}' = 0.0013 = [^{15}0.63\ \mu M\ /(^{15}0.63 + {}^{14}487\ \mu M)]^{..}$ | ≤0.8 |

Ambient background concentrations for $^{14}$N$_2$ and $^{14}$N$_2$O in both our $^{15}$N$_2$ and $^{15}$N$_2$O treatments were ~487 μM and 0.01 μM, respectively. We added both $^{15}$N$_2$ and $^{15}$N$_2$O at 9 μM (>98 atom % $^{15}$N), resulting in initial $^{15}$N labelling of the $^{15}$N$_2$ and $^{15}$N$_2$O pools of 0.018 and 0.98 ($F_{N2}$ and $F_{N2O}$, respectively). If $^{15}$N$_2$O assimilation was indirect, and $^{15}$N$_2$O was first reduced to $^{15}$N$_2$, then at most 0.63 μM $^{15}$N$_2$ would have been produced and $F_{N2}'$ would have been ≤0.0013. Accordingly, the absolute upper threshold for indirect $^{15}$N$_2$O fixation – in proportion to that directly with $^{15}$N$_2$ ($F_{N2}$) – would have been 0.8 i.e., [(0.0013/0.018)×11.5] nmol N g$^{-1}$ d$^{-1}$, which is far lower than our measured rates for $^{15}$N$_2$O assimilation (5.3 nmol N g$^{-1}$ d$^{-1}$, on average, Fig. 2b).
*With the predicted maximum $^{15}$N-labelling of the N$_2$ pool ($F_{N2}'$) resulting from the maximum reduction of $^{15}$N$_2$O to $^{15}$N$_2$.
$^{..}$Where $^{15}$ and $^{14}$ denote the $^{15}$N and $^{14}$N species, respectively.

community or a sub-set, using a long-term incubation with N$_2$O-enriched biomass.

N$_2$O reduction was most rapid during the first 3 days of the incubation and started to decline after the increase in total inorganic nitrogen (TIN, sum of NO$_3^-$, NO$_2^-$ and NH$_4^+$) from 0.44 μM to 0.76 μM and (Fig. 5a and Supplementary Fig. 7a). We terminated the incubation after 25 days when oxygen production from photosynthesis started to decline (Fig. 5a and Supplementary Fig. 8b) and characterised the abundance and structure of the *nifH* community.

We first tested whether N$_2$O reduction was related to the abundance of the whole *nifH* community (copy numbers of *nifH* per g wet biomass) but found no relationship ($p = 0.21$, $F = 1.74$, d.f.=18). We then tested whether N$_2$O reduction was related to a sub-community by looking for any changes in diversity or composition of the *nifH* community over the 25-day incubation. Our primers amplified 894 well-represented OTUs ( > 20 reads in at least 3 samples) of which only 227 were identified as *nifH* OTUs (*see* Methods and Supplementary Figs. 9 and 10). However, neither the diversity ($p = 0.81$, $t = 0.24$, d.f. = 17.5, two-sided *t*-statistic tested on the means of Shannon index) nor the composition ($p = 0.99$, PERMANOVA, *see* also Supplementary Fig. 11 for unchanged *nifH* community composition at 3, 10 and 25 days) of the overall *nifH* community changed significantly during the 25-day incubation.

As an alternative, we used redundancy analysis (RDA) to ordinate the relative abundances of *nifH* OTUs and the initial rates (i.e., in 10 samples up to day 3) of N$_2$O reduction to identify any likely N$_2$O fixing candidates (Fig. 5b). Since N$_2$O reduction may be mediated by a subset of the whole *nifH* community, we relaxed our definition of a well-represented OTU to include <20 reads in at least 3 samples (see Supplementary Fig. 9) which retained 72 out of 227 *nifH* OTUs. Of those 72 OTUs, the relative abundance of 22 were positively correlated with N$_2$O reduction i.e., in the ordination their arrows pointed in a similar direction to the arrow for N$_2$O reduction. The positive correlations for the 22 OTUs, including 15 Cyanobacteria and 7 Proteobacteria, were further explored by visualising their relative abundance in each biomass sample in rank order of increasing rate of N$_2$O reduction (Fig. 5c). Among the 15 Cyanobacterial OTUs, *Pegethrix*-like OTU392 and OTU394 (100% identical protein sequence to *Pegethrix*) and the *Fischerella*-like (>99% identical) OTU396 appeared to not only be more common, but they were also more strongly correlated with the initial rates of N$_2$O reduction. While OTU412 and OTU389 were also identical to *Pegethrix* they were either relatively rare or less-well correlated, respectively. Despite the two *Methylomonas*-like (>99 % identical) Proteobacterial OTU444 and OTU462 being less common than the Cyanobacterial candidates, exisiting in only four samples, their higher relative abundances coincided with higher rates of N$_2$O reduction. Moreover, combinations of the five strongest (OTUs 392, 394, 396, 444, 462) N$_2$O fixing candidates were not only present in the ten samples used to determine the initial rates of N$_2$O reduction but all 40 samples enriched with N$_2$O for our 25-day incubation (Supplementary Fig. 9).

## Discussion

In ecosystems with limited fixed nitrogen (e.g. inorganic NO$_2^-$, NO$_3^-$, NH$_4^+$), primary production is tightly coupled to N-fixation – typically recognised to be N$_2$ gas (Eq. 2). The undersaturation in N$_2$O reported here means that the reduction of N$_2$O was greater than its rate of delivery either from the atmosphere or biological sources in the ponds, which shows that these N-limited ponds were overall sinks for N$_2$O, including direct N$_2$O fixation (Eq. 1).

Others have argued for direct N$_2$O fixation on the premise that if N$_2$ production was not detected in incubations with N$_2$O, then N$_2$O fixation was direct[5,30]. Here, besides not detecting $^{15}$N$_2$ production in 76% of our incubations (Fig. 2b), our disproportionate fixation of N from $^{15}$N$_2$O relative to that measured with $^{15}$N$_2$ provides more substantive evidence for direct N$_2$O fixation (Table 1). Direct N$_2$O fixation represents an alternative to the only widely recognised sink for N$_2$O – namely denitrification[6,29]. In addition, our estimation of in situ N$_2$O fixation helps to rationalise the undersaturation and resultant flux of N$_2$O into our ponds (see Supplementary text 2). Further, as the scale of N$_2$O undersaturation in our ponds (Fig. 1a) is in line with many other studies also reporting undersaturation in N$_2$O in freshwaters (typically ~70–100%)[10,16–18,21], this indicates that direct N$_2$O fixation could explain the unaccounted for N$_2$O undersaturation in many freshwaters[8–10,16–21].

We can see similar seasonal trends to what we report here in previous accounts of N$_2$O undersaturation. For example, boreal lakes, ponds and rivers show undersaturation in N$_2$O which is strongest at coldest temperatures[19]. In Boreal peatlands, N$_2$O was undersaturated mostly in spring, increasing to maximum oversaturation in summer, then decreasing to near equilibrium in autumn[46]. From the same study, soils acted as net N$_2$O sources at higher temperatures, while most N$_2$O sinks occurred below 13 °C[46]. In the surface waters of the Baltic Sea, N$_2$O was most undersaturated in winter (December), but was oversaturated in summer and autumn[11]. However, these studies generally lacked a clear explanation for the occurrence and the temperature dependence of N$_2$O undersaturation, whereas we now offer an explanation.

Our findings demonstrate different temperature dependencies for N$_2$ and N$_2$O fixation. This difference in N$_2$ versus N$_2$O is supported not only by the opposing seasonal patterns in N$_2$ and N$_2$O saturation in our ponds, but also by the experimentally determined different temperature sensitivities for the assimilation of N$_2$ and N$_2$O by biomass in our incubations. Moreover, the results from our incubations support the seasonal patterns in N$_2$ and N$_2$O saturation in the ponds – with the higher rates of N$_2$O reduction in incubations in winter than in summer, matching the stronger N$_2$O undersaturation in the ponds in winter, and the elevated temperature effect on N$_2$ assimilation agrees with that for N$_2$ saturation in our ponds. Here the apparent lack of temperature sensitivity of N$_2$O fixation (Fig. 4) suggests that the N-fixing communities may be strongly adapted to substrate limitation (Supplementary Fig. 6), with dissolved N$_2$O typically at ~10 nM in the ponds compared to ~490 μM for N$_2$. This strong kinetic effect of substrate availability on N$_2$O fixation has also been reported in incubations with surface

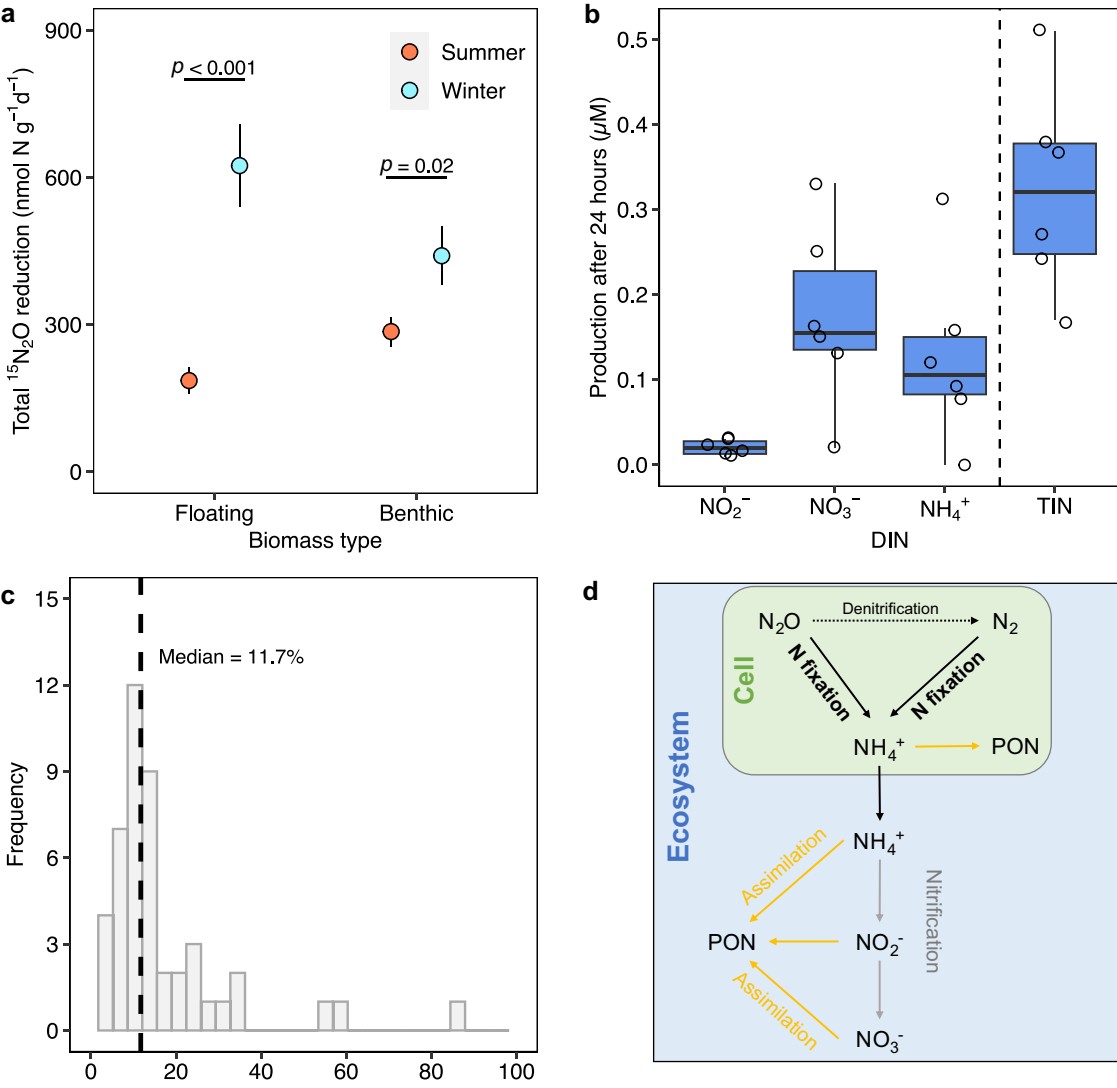

**Fig. 3 | The fate of total $^{15}N_2O$ reduction in biomass. a** Total $^{15}N_2O$ reduction was highest in winter compared to summer in both floating ($p < 0.001$, two-sided) and benthic biomass ($p = 0.02$, two-sided). Data plotted are means ± s.e. from 95% (2.5% to 97.5% percentiles) of the dataset for summer and winter where $n = 180$ and $n = 160$ incubations in summer and winter, respectively (two months for summer, three months for winter). **b** Production of dissolved inorganic nitrogen in $^{15}N_2O$ incubations. TIN: Total inorganic nitrogen. $n = 6$ incubations for biomass from 6 ponds. Each box shows the 25th to 75th percentiles, horizontal lines the median, and whiskers extend 1.5 times the interquartile range. **c** Distribution of the ratio of $^{15}NO_x^-$ production to total $^{15}N_2O$ reduction, dashed line is the median. Data in **c** are from December, 2020, when the highest rate of total $^{15}N_2O$ reduction was measured ($n = 47$ incubation bottles, 5 temperatures with 2 biomass types). **d** Simplified diagram showing possible pathways for $N_2O$ reduction in relation to canonical $N_2$ fixation.

seawater[23] and pure culture of marine cyanobacteria *Trichodesmium* sp[5]. Similar stronger limitation of activity by substrate over temperature is also recognised in other autotrophs such as the methane oxidising methanotrophs[47,48].

The contrasting temperature sensitives of $N_2O$ and $N_2$ fixation are probably associated with the energetic advantage of using $N_2O$ instead of $N_2$ as a N-substrate for N-fixation[49]. Here we compiled data for studies measuring $N_2$ fixation in both aquatic and terrestrial ecosystems (Supplementary Fig. 12 and the references cited therein) which clearly shows $N_2$ fixation increases at higher temperatures. On the other hand, as dissociating the N bond in $N_2O$ only requires about half of the energy compared to $N_2$ (refs. 49,50), $N_2O$ should be easier to fix at colder temperatures and a higher proportion of total N fixation could be dependent on $N_2O$ in the cold. For example, with our pond biomass the fraction of total N-fixation coupled to $N_2O$ at 6°C was 26% higher than that at 25°C (Fig. 4a). This energy saving offered by fixing $N_2O$ in the cold might explain why $N_2O$

undersaturation in our ponds was strongest during the colder months and may also explain the undersaturation reported in cold, boreal environments and Baltic Sea[11,19,46].

In addition, in northern latitudes cold temperatures typically occur alongside lower light and the -18% energy saving from fixing $N_2O$ (Eq. 1) compared to $N_2$ (Eq. 2) could provide an over-wintering strategy for a subset of the photosynthetic community. Besides photosynthesis, some chemosynthetic microbes could also benefit. For example, some *Methylomonas* spp. are known to fix $N_2$ and were identified here amongst our potential $N_2O$ fixing candidates. Some 32% of the energy yielded from oxidising $CH_4$ to $CO_2$ can be expended on assimilating $CH_4$ into biomass (equation S6, Supplementary Text 1) with another 56% needed to fix $N_2$ compared to 47% for $N_2O$. Given that methane concentrations are lowest in our ponds in winter[39] - and more widely methane production is known to be tightly coupled to primary production in spring and summer[51] - fixing $N_2O$ could offer an advantage over $N_2$ when resources are limited.

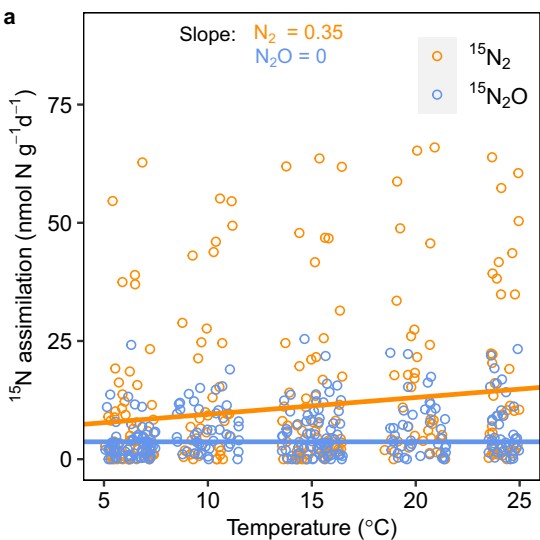

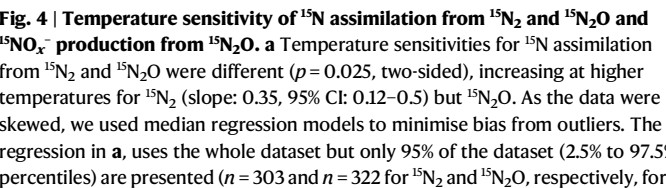

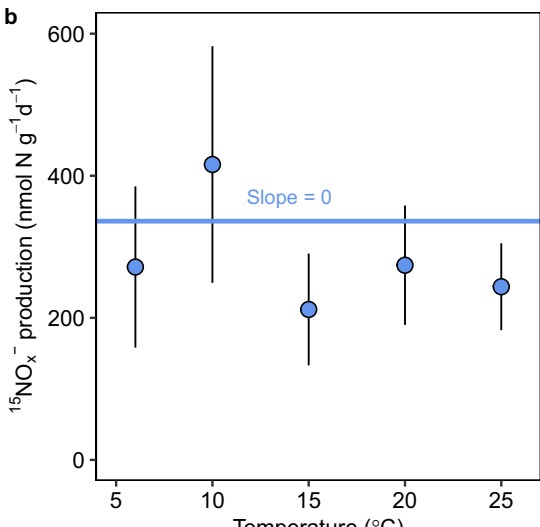

**Fig. 4 | Temperature sensitivity of $^{15}$N assimilation from $^{15}$N$_2$ and $^{15}$N$_2$O and $^{15}$NO$_x^-$ production from $^{15}$N$_2$O. a** Temperature sensitivities for $^{15}$N assimilation from $^{15}$N$_2$ and $^{15}$N$_2$O were different ($p = 0.025$, two-sided), increasing at higher temperatures for $^{15}$N$_2$ (slope: 0.35, 95% CI: 0.12–0.5) but $^{15}$N$_2$O. As the data were skewed, we used median regression models to minimise bias from outliers. The regression in **a**, uses the whole dataset but only 95% of the dataset (2.5% to 97.5% percentiles) are presented ($n = 303$ and $n = 322$ for $^{15}$N$_2$ and $^{15}$N$_2$O, respectively, for 5

months, 2 biomass types). **b** similarly, $^{15}$NO$_x^-$ production from $^{15}$N$_2$O was also invariant to temperature. Data in **b** are from December, 2020, when total $^{15}$N$_2$O reduction was highest ($n = 47$ incubations, 5 temperatures, 2 biomass types), blue line is a first-order linear regression, mean ± s.e. As the temperature sensitivity of $^{15}$N assimilation and $^{15}$NO$_x^-$ production was consistent between floating and benthic biomass, the data in **a** and **b** have been pooled for both biomass types.

The concept that N$_2$ fixation in general is routinely supressed by inorganic N (>-1 μM) has been revised[52]. For example, while there are numerous examples of N$_2$-fixing activity being suppressed by inorganic nitrogen in pure cultures of *Trichodesmium* spp., others have reported its activity in euphotic ocean waters (whole water and *Trichodesmium* spp. isolates) with ~5 μM to 21 μM NO$_3^-$ (refs. 52,53), which may indicate short-term tolerance to NO$_3^-$. In contrast, the N-fixing community in our ponds is exposed to chronic, year round inorganic nitrogen starvation (i.e., <1 μM). Our 25-day incubation showed N$_2$O reduction declined after an increase in inorganic nitrogen (NO$_3^-$ and NH$_4^+$) to ~0.8 μM (Fig. 5a), indicating a low-threshold concentration for inorganic nitrogen that apparently limits N$_2$O fixation. Such a threshold also reflects the co-occurance of N$_2$O undersaturation (Fig. 1a) with <1 μM inorganic nitrogen throughout the year in our ponds (Fig. 1f). In contrast, N-rich ecosystems generally act as N$_2$O sources[8,25], while N$_2$O sinks - mediated through N$_2$O fixation – are likely to be found in pristine, cold ecosystems[19,21,22,26–28].

To date, it is not clear which microorganisms are responsible for N$_2$O fixation in natural ecosystems. A few studies have reported N$_2$O fixation in sea water[5,23] and soybean root nodules[30], but only one study, on pure cultures of the marine cyanobacteria *Trichodesmium* sp. and *Crocosphaera* sp., has related the *nifH* gene to N$_2$O fixation[5]. Since being set up in 2005, our nitrogen-limited ponds have developed diverse diazotroph communities[42], comparable to those in estuaries[54], freshwater[55] and seawater[56]. Here we set out to link cause to effect by attempting to enrich N$_2$O fixing candidates but failed to detect any changes in the total *nifH* community after 25 days enrichment with N$_2$O (Supplementary Fig. 11). This might have been due to the decline in rate of N$_2$O reduction, as a result of the parallel accumulation in inorganic nitrogen, or longer incubations being needed to detect any change in the total *nifH* community that is likely slow-growing. In our ordination analysis, five OTUs emerged as the strongest potential candidates for N$_2$O fixation i.e., their relative abundance is at least correlated with N$_2$O reduction activity. Of these, one is identical to *Fischerella*, a common diazotroph in freshwater[55] and seawater[57] and also identified in seawater undersaturated in N$_2$O[23] and another two being identical to *Pegethrix*[58], a newly identified genera of filamentous Cyanobacteria. In

addition, proteobacterial methanotrophic diazotrophs have been related to N-fixing activity in N-limited freshwaters[59], and relatives of our two *Methylomonas*-like candidates are known to grow on N$_2$ as their sole nitrogen source[60]. Whether or not our candidate N$_2$O-fixers are responsible for the widely reported undersaturation in N$_2$O in natural waters needs characterising directly but our first attempt here at least suggests N$_2$O fixation is likely mediated by a subset of diazotrophic communities.

In addition, our work shows that N$_2$O fixation can occur in an abundance of N$_2$ i.e., against the high natural N$_2$ background. This indicates that N$_2$O fixation could happen in natural ecosystems replete in N$_2$ and provides further insight into the communities responsible for N$_2$O fixation. For example, *nifH* communities could fix N$_2$ and N$_2$O randomly, with the ratio of N$_2$O to N$_2$ fixation being simply proportional to the relative availability of N$_2$O to N$_2$. However, the distinct seasonal patterns we measured for N$_2$ and N$_2$O undersaturation (Fig. 1), coupled to disproportionate rates of N$_2$O fixation (Fig. 2) and higher proportion of N$_2$O fixation at colder temperatures (Fig. 4a) - all indicate that a specialised subset of the *nifH* community (Fig. 5) likely favoured N$_2$O over N$_2$ at colder temperatures in support of our *nifH* ordination analysis.

To put our estimates of N fixation into an ecological context, we compared estimates of the N$_2$ flux (Supplementary Text 2) with former estimates of gross primary production (GPP) in the ponds[40]. For example, the average net N$_2$ flux into the ponds was 3934 μmol N$_2$ m$^{-2}$ d$^{-1}$, which, assuming Redfield ratios of 106:16 for C:N, could sustain GPP of 52,126 μmol C m$^{-2}$ d$^{-1}$ and which is comparable to GPP measured previously of 51,488 to 70,792 μmol C m$^{-2}$ d$^{-1}$ (ref. 40). Moreover, the seasonal dynamics in N$_2$ flux in our study also matched that of GPP reported previously[40], with both peaking in the summer (Fig. 1b). In contrast, while the flux of N$_2$O is comparatively minor (-0.03 %) in terms of supporting GPP, it is great enough to maintain a strong sink for N$_2$O.

To date, denitrification in either anoxic or oxic-to-anoxic transitioning waters is still the only widely recognised sink for N$_2$O[6,7,29]. Here, as an alternative to denitrification, direct N$_2$O fixation can rationalise the undersaturation in N$_2$O in our ponds and could also explain the various unaccounted for N$_2$O sinks – of similar magnitude – reported

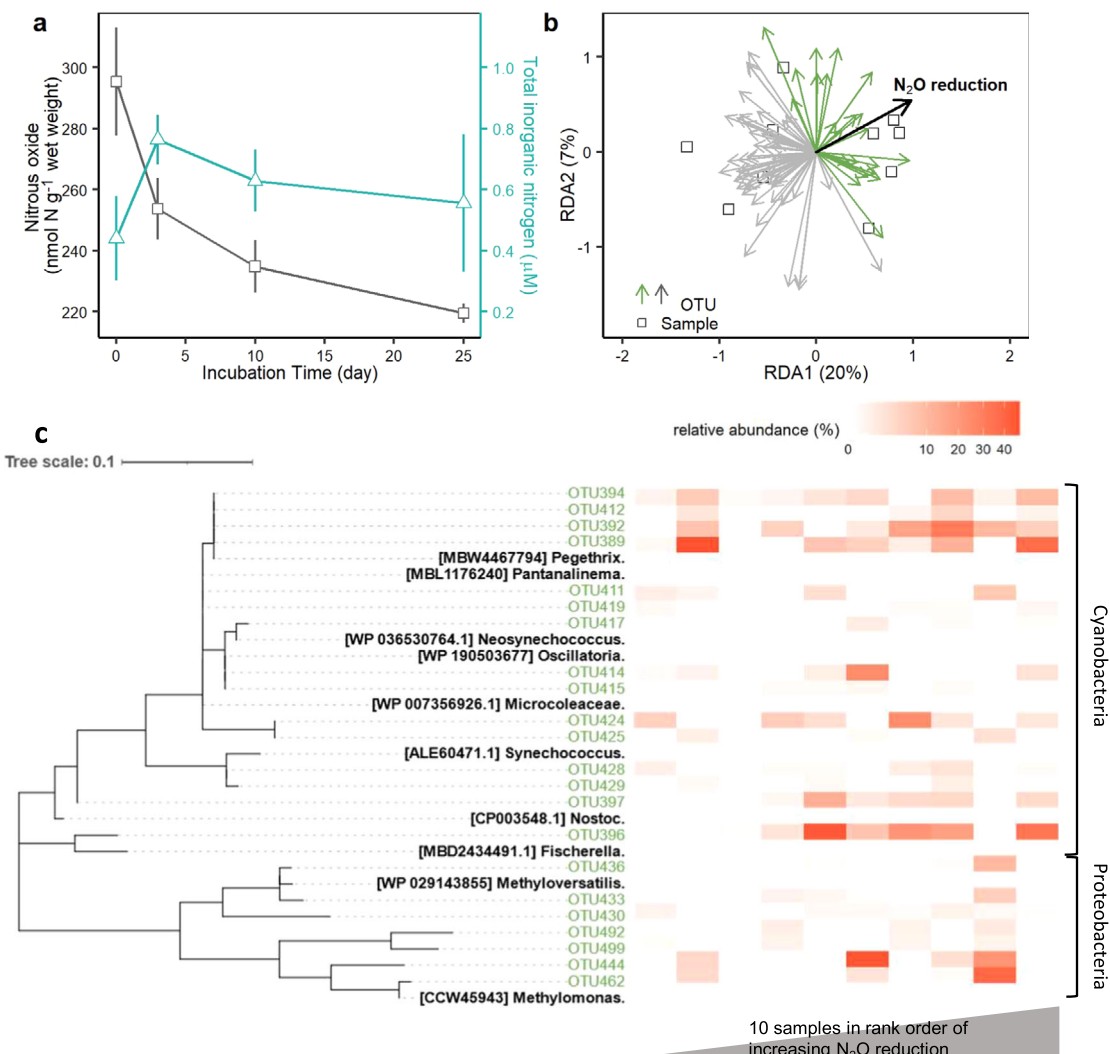

**Fig. 5 | Biomass *nifH* community in relation to N₂O reduction. a** Biomass incubated for 25 days reduced 76 nmol N₂O-N g⁻¹ (dry weight) on average (*n* = 40 incubations for biomass enriched with N₂O) with activity peaking before inorganic nitrogen accumulated. Data plotted are means ± s.e. **b** Redundancy analysis (RDA) revealed positive correlations between the initial rates of N₂O reduction and the relative abundance of 22 *nifH* OTUs, including 15 Cyanobacterial OTUs and 7 Proteobacterial OTUs (arrows in green). **c** Heat-map (white to dark red) of the relative abundance of the 22 *nifH* OTUs (in green), identified in **b**, in samples (columns, *n* = 10) in rank order of increasing rate of N₂O reduction. *n* = 10 in **b** and **c**. N₂O reduction is presented as a black arrow in **b** and as a grey ascending triangle in **c**.

in natural, pristine waters[10,17,19,21,22,28,46]. As N₂O undersaturation is favoured in the cold, rising temperatures could erode this natural sink for such a potent climate-gas.

## Methods

### Nutrient analysis

Temperature and O₂ were measured in each pond using HQD portable metre (Hach). Samples of water for nutrient analysis were filtered (0.45 μm PES, 25 mm, pre-washed with deionized water) into Falcon tubes, kept cool and frozen at −20 °C back in the laboratory. Samples were thawed overnight at 4 °C and analysed by standard wet-chemistries for NO₂⁻, NO₃⁻, NH₄⁺ and SRP on an autoanalyzer (San⁺⁺, SKALAR Analytical B.V.)[61] against certified reference materials, traceable to NIST. The limits of detection were 0.05 μM and 0.1 μM for NO₂⁻ and NOₓ⁻ (NO₂⁻ + NO₃⁻), respectively, 0.2 μM for NH₄⁺ and 0.05 μM for SRP. SRP and total inorganic nitrogen (TIN, NO₃⁻ + NO₂⁻ + NH₄⁺) below detection limits were omitted from any calculations.

### Dissolved N₂ and N₂O in the ponds

For dissolved N₂ and N₂O analyses, water samples were taken carefully at mid-water-depth (~20 cm from the surface) from each pond using a 60 mL syringe and tubing. Five gas-tight vials (12 mL Exetainer, Labco, two vials for N₂O and three for N₂) for each pond were allowed to overflow three times, preserved with ZnCl₂ (50 μL of 50% w/v)[62], closed and mixed by hand. Extra pond water samples for reference N₂ saturation were collected and preserved along with samples of air.

In a temperature-controlled laboratory at 22 °C, references for N₂ saturation were prepared by equilibrating the pond water with the laboratory air, and then water and air samples were collected as for the field samples. Helium headspaces (2 mL 99.999% purity) were created in all sample and reference vials, followed by 24 h equilibration on an orbital shaker (SSL1, Stuart) in the same laboratory and all vials weighed to determine the exact volume of headspace and water.

For N₂O, 100 μL of sample headspace was injected by an auto-sampler into a gas chromatograph fitted with a μECD (Agilent Technology UK Ltd., South Queensferry, UK) along with air samples using conditions described previously[63]. Calibration was performed against known concentrations of N₂O from a NOAA standard (traceable to the SI unit "amount of substance fraction") at 359.73 ppb or 120 ppb and 1.04 ppm and 96 ppm from BOC, UK, cross-calibrated to the NOAA standard. The precision for N₂O concentration was 2% (coefficient of variation, *n* = 10). The total concentration of N₂O in each vial was

calculated using solubility coefficents[64] as described before[63] and the degree of over- or under-saturation calculated by comparison to the expected concentration of $N_2O$ for pond water at equilibrium with the atmosphere (see Supplementary Fig. 1).

For $N_2$ analysis, we used the published $N_2$:Ar method[65]. 100 µL of headspace was injected by an autosampler into an elemental analyzer, (Flash EA 1112 series, Thermo Finnigan) to remove $O_2$ by the hot-copper reduction, before passing to a continuous flow isotope ratio mass spectrometer (CF-IRMS, Delta V Plus, Thermo Finnigan). Throughout each run, air samples were analysed to correct for drift and the expected concentrations for $N_2$ or Ar in the headspace ($C_{hs}$) calculated using the solubility of $N_2$ and Ar for air at both field ($K_{field}$) and laboratory ($K_{lab}$) temperature[66]:

$$C_{hs} \times V_{hs} + C_{hs} \times K_{lab} \times V_{aq} = C_{field} \times K_{field} \times V_{aq} \qquad (4)$$

Where $V_{hs}$ and $V_{aq}$ are the volumes of headspace and water in a vial and $C_{hs}$ and $C_{field}$ the concentration of either gas in the headspace or field air, respectively. The saturation of $N_2$ in the samples was then derived by comparing the measured to expected ratio of $N_2$ to Ar in the samples to that in the pond water ref. [67]:

$$N_2 \text{Saturation}(\%) = \left(\frac{N_2/\text{Ar measured}}{N_2/\text{Ar expected}}\right)_{\text{Sample}} \bigg/ \left(\frac{N_2/\text{Ar measured}}{N_2/\text{Ar expected}}\right)_{\text{reference}} \times 100 \qquad (5)$$

Precision for the ratio of $N_2$ to Ar for triplicate reference water and air standards was 0.1% and 0.05% (coefficient of variation), respectively. We also tested the effect of calculating $N_2$ saturation with different references, with the ratio for deionized versus pond water being 99.7% and 99.82%, on average ($n = 20$ and $n = 8$, respectively).

### Biomass incubations to characterise $N_2$ and $N_2O$ fixation

Two types of biomass were collected from the ponds for the routine (24-h) incubations (Supplementary Fig. 3) and see below for the 25-day incubation. Floating assemblages on the ponds, comprising *Oedogonium* spp. and microorganisms attached to the filaments (Supplementary Fig. 3b) and green or yellow benthos assemblages (Supplementary Fig. 3c), sampled avoiding the sandy sediments beneath. Samples of biomass were collected in sterile Falcon tubes (50 mL) and transported back to the laboratory in a cool box and stored overnight at 15 °C before preparing the incubations. Despite the ponds being low in dissolved inorganic nitrogen, we standardised nutrient concentrations in the incubations by using an artificial pond water medium devoid of fixed N comprising: $CaCl_2$ (0.5 mM), KCl (1 mM), $MgSO_4$ (0.25 mM), $KHCO_3$ (0.7 mM) and $NaHCO_3$ (0.5 mM) in deionized water. P was added to 0.08 µM of $NaPO_4$, based on measured SRP concentrations in the ponds.

### Incubations of biomass with $^{15}N_2$ and $^{15}N_2O$ tracers

Floating or benthic biomass was weighed (~3 g wet weight) into 12 mL gas-tight vials and, for the $^{15}N_2O$ treatment, filled with oxygen-saturated artificial pond water and closed. 100 µL of water was replaced by $^{15}N_2O$ stock solution (see below) with a gastight syringe. For the $^{15}N_2$ treatment, each vial was filled with 10 mL of oxygen-saturated artificial pond water and 2 ml of $^{15}N_2$ stock and closed (see below). All incubations were prepared without a headspace to ensure $^{15}N$-substrate concentrations were the same under different temperatures. Parallel controls were prepared in the same way without either $^{15}N$-gas.

$T_0$ (Time zero) vials were killed with 200 µL of 50% (w/v) formaldehyde immediately and the remainder incubated in temperature-controlled orbital incubators (SI500, Stuart at 50 cycles min$^{-1}$) at 6, 10, 15, 10 and 25 °C on a 12 h light/12 h dark cycle for 24 h. Time final ($T_f$)

samples were killed as above, brought to 22 °C, helium headspaces created in all vials, and all allowed to equilibrate for 24 h, as above.

### Stock solutions for $^{15}N_2$ and $^{15}N_2O$ additions to the biomass incubations

To avoid the recognised equilibration problems with the $^{15}N_2$ "bubble method", especially during short-term incubations[68], we first made an aqueous $^{15}N_2$ stock with the artificial pond water. 200 mL of artificial media were injected into a 0.5 L gas sampling bag along with 40 mL $^{15}N_2$ gas (98% atom % $^{15}N$, Sigma-Aldrich) and allowed to equilibrate for 24 h while gently rocking. $^{15}N_2O$ stock solutions were prepared by replacing 3 mL of water with $^{15}N_2O$ gas (98% atom % $^{15}N$, Cambridge Isotope Laboratories, Inc.) in a 50 mL sealed serum bottle.

The solubility of $N_2$ is low and to maximise $^{15}N_2$ labelling we added a relatively large amount (~2 ml) of $^{15}N_2$ stock solution to the 12 mL vials (~16% v/v). To keep the dissolved nutrients and gases at background levels in the $^{15}N_2$ treatments -and the same as in the controls and $^{15}N_2O$ treatment- we used artificial freshwater medium instead of deionized water for preparing $^{15}N_2$ stocks. In contrast to $N_2$, $N_2O$ is highly soluble, with only ~100 µL of $^{15}N_2O$ stock being needed in each treatment (~0.8% v/v) to reach the comparable concentration of $^{15}N$-$N_2$ addition (~10 µM). $^{15}N_2O$ and $^{15}N_2$ stocks were prepared fresh before each experiment and their respective dosages tested by spiking controls.

### Characterising total $^{15}N_2O$ reduction

The concentration of $^{15}N_2O$ in the samples was measured on a CF-IRMS (Delta V Plus, Thermo-Finnigan) with an automated trace gas pre-concentrator (PreCon, Thermo-Finnigan)[63]. A sub-sample from the headspace of each sample was transferred to a 12 mL air-filled gas-tight vial. The high $^{15}N$-labelling of $^{15}N_2O$ (on average, 97.7% of $^{46}N_2O$) in the $^{15}N_2O$ treatment meant only a small aliquot of sample (10 µL) was needed to keep the signal within the measurable range of $^{46}N_2O$. Mass-to-charge ratios were measured for $m/z$ 44, 45 and 46 and the concentration of $N_2O$ determined by calibration against known amounts (0.02–2 nmol) of natural abundance $N_2O$ (96 ppm $N_2O$ standard, BOC, UK)[63]. Note, whether the mass spectrometer is calibrated with high purity $^{15}N_2O$ or natural abundance $N_2O$, the signal-to-mole ratio is constant (Supplementary Fig. 13). Here the concentration of total $^{15}N_2O$ is expressed as $^{15}N_2O = {}^{45}N_2O + 2 \times {}^{46}N_2O$ and the reduction of $^{15}N_2O$ calculated by subtracting $^{15}N_2O$ concentrations in $T_f$ samples from $T_0$ samples.

### Characterising any dissimilatory reduction of $^{15}N_2O$ to $^{15}N_2$

Any production of $^{15}N_2$ in the $^{15}N_2O$ treatments was measured by the CF-IRMS (Delta V Plus, Thermo Finnigan), after bypassing the copper reduction step to avoid reduction of $^{15}N_2O$ to $^{15}N_2$ (ref. [69]). The concentration of total $N_2$ was calculated from the solubility of $N_2$ (ref. [66]) and the signal of total $N_2$ mole masses i.e., $m/z$ 28, 29 and 30 in the samples and air standards[70]. Drift in $m/z$ 30 was corrected by inserting air standards for every 10 samples. Changes in the concentration of $^{15}N_2$ ($\Delta^{15}N_2$, nmol N d$^{-1}$) were calculated by the excess $^{15}N_2$ in $^{15}N_2O$ treatments compared to the controls, where $\Delta^{15}N_2 = \Delta^{29}N_2 + 2 \times \Delta^{30}N_2$. The limit of detection for $\Delta^{15}N_2$ in the incubations is ~0.14 µM.

### Characterising assimilation of $^{15}N_2O$ or $^{15}N_2$ into biomass

After all the gas measurements, samples were centrifuged and the supernatants filtered (as above). The remaining biomass was dried, homogenised and sub-samples weighed into tin caps (6 × 4 mm, Elemental Microanalysis) for elemental analysis as described previously[41]. The level of $^{15}N$ enrichment in biomass incubated with either $^{15}N_2$ or $^{15}N_2O$ was then calculated by the difference in excess $^{15}N$ atom % relative to the controls, where excess $^{15}N$ atom % is the difference in $^{15}N$

atom % between $T_0$ and $T_f$ in the 24-h incubations:

$$^{15}N \text{ enrichment} = (excess^{15}N \text{ atom\%})_{Treatment} - (excess^{15}N \text{ atom\%})_{Control} \tag{6}$$

Rates of $^{15}N$ assimilation (nmol $^{15}N$ g$^{-1}$ day$^{-1}$) by biomass into particulate organic nitrogen (PON) were calculated as:

$$^{15}N \text{ assimilation rate} = PON \times {^{15}N} \text{ enrichment}/(\Delta t \times dw) \tag{7}$$

Where PON is particulate organic nitrogen in a sample of biomass, $\Delta t$ is the incubation time (24 h), and dw the dry weight (g).

## Characterising $^{15}N_2O$ fixation

As total $^{15}N_2O$ reduction includes both assimilatory $^{15}N_2O$ fixation and dissimilatory $^{15}N_2O$ reduction to $^{15}N_2$, total $^{15}N_2O$ fixation can be calculated by subtracting $^{15}N_2$ production from total $^{15}N_2O$ reduction:

$$\text{Total}^{15}N_2O \text{ fixation} = \text{Total}^{15}N_2O \text{ reduction} - {^{15}N_2} \text{ production} \tag{8}$$

Where total $^{15}N_2O$ fixation includes $^{15}N_2O$ assimilated into biomass, as well as any fixed $^{15}N_2O$ present in the pond water medium as dissolved inorganic nitrogen ($^{15}DIN$, e.g., $^{15}NH_4^+$, $^{15}NO_2^-$ and $^{15}NO_3^-$):

$$\text{Total}^{15}N_2O \text{ fixation} = {^{15}N_2O} \text{ assimilation} + {^{15}DIN} \text{ production} \tag{9}$$

We characterised any $^{15}DIN$ production coupled $^{15}N_2O$ fixation by measuring $^{15}NO_x^-$ (i.e., $^{15}NO_3^- + {^{15}NO_2^-}$) with sulfamic acid[62], testing for any effect of formaldehyde on the $^{15}NO_x^-$ assay (Supplementary Fig. 14). Due to the high $^{15}N_2O$ background, it was not possible to measure any $^{15}NH_4^+$ from $^{15}N_2O$ using the sensitive sodium-azide assay as it converts $^{15}NO_2^-$ to $^{15}N_2O$. Further, as the formaldehyde preservative interferes with the colorimetric $NH_4^+$ assay, we did additional incubations in October 2021 without formaldehyde, following the exact incubation procedure described above. Here, samples for DIN were immediately centrifuged and frozen at $-20\,°C$ while parallel samples for gases were treated as above. Concentrations of DIN in controls and $^{15}N_2O$ treatments were measured by the automated wet-chemistry autoanalyzer (see 'Nutrient analysis' in Methods), while changes in $N_2O$ concentrations were measured by GC/μECD[63].

## $nifH$ communities in relation to $N_2O$ reduction over a 25-day incubation

To characterise any $nifH$ communities potentially involved in $N_2O$ fixation, we incubated floating biomass from the ponds with excess $N_2O$ for as long as possible, looking either for changes in the $nifH$ community or relationships between particular $nifH$ families and $N_2O$ reduction. Floating biomass was collected from 10 ponds in May 2021 (as above) and once back in the laboratory kept in a temperature-controlled room at 15°C overnight. The next day, 7 g wet biomass was transferred into 70 mL serum bottles ($n = 80$, 8 serum bottles per pond), filled with water and sealed. $N_2O$ stock solution (600 μL as above) was injected into half of the serum bottles, while venting water through a needle, to create an initial $N_2O$ concentration of ~10 μM and the remaining serum bottles left unamended as controls. Temperature-controlled incubations were carried out on a 12 h:12 h light/dark cycle as above. A total of 20 serum bottles (10 with $N_2O$ and 10 controls) were sacrificed after 0, 3 and 10 days of incubation, respectively, while the last 20 serum bottles were incubated until the daily maximum in oxygen started to decline. Oxygen was measured with optical sensors (OXSP5, FireSting®, Pyro Science GmbH, Germany) at ~2-hourly intervals after lights and the incubations terminated on day 25 when daily maximum oxygen started to decline (Supplementary Fig. 8).

Sub-samples of water were transferred from each serum bottle into a 3 ml gas-tight vial (Exetainers, Labco) after 0, 3, 10 and 25 days of incubation, fixed with 50 μl formaldehyde, sealed and stored at room temperature. After creating a helium headspace $N_2O$ concentrations were measured by GC/μECD (as above). The remaining water was filtered and frozen at $-20\,°C$ for later quantification of $NO_3^-$, $NO_2^-$ and $NH_4^+$, as above. Biomass was frozen at $-20\,°C$ until DNA extraction (June 2021) from ~0.5 g of wet biomass (DNeasy PowerSoil kit, Qiagen) as per the manufacturer's instructions.

## $nifH$ gene abundance (qPCR) and library preparation

Gene abundance of $nifH$ was determined using qPCR with IGK3/DVV (forward, 5'-GCIWTHTAYGGIAARGGIGGIATHGGIAA-3'; reverse, 5'-ATIGCRAAICCICCRCAIACIACRTC-3')[71] using a CFX384 Touch Real-Time PCR (Bio-Rad) in 10 μL reactions containing 5 μL SensiFAST SYBR No-ROX mastermix (Meridian Bioscience), 0.8 μL of each primer (10 μM), 0.8 μL DNA template and 2.6 μL molecular biology quality water (MBQW). The qPCR programme was 98 °C (3 min) then 40 cycles of 98 °C (15 s), 58 °C (60 s), 72 °C (60 s). Standard curves ($10^5$ to $10^8$ copies per μL) were prepared from plasmid DNA containing $nifH$ and product specificity confirmed by endpoint melt curve analysis.

A three step PCR was used to prepare the $nifH$ library[72]. $nifH$ was amplified using IGK3/DVV in 10 μL of MyTaq Red Mix (Bioline), 0.8 μL of each primer (10 μM), 0.8 μL of DNA template and 7.6 μL MBGW on a T100 Cycler (Bio-Rad) at (1) 94 °C (5 min); (2) 36 cycles of 94 °C (30 s), 57 °C (45 s), 72 °C (30 s); (3) 72 °C (10 min). These PCR products were then re-amplified with IGK3/DVV appended with overhang MiSeq adaptors in 25 μL containing 12.5 μL of MyTaq Red Mix (Bioline), 1 μL of each primer (10 μM), 1 μL of amplicons from the first step as template and 9.5 μL of MBGW. The PCR programme was: (1) 94 °C (4 min); (2) 12 cycles of 94 °C (30 s), 57 °C (45 s), 72 °C (30 s); (3) 72 °C (7 min). PCR products were cleaned using AMPure XP beads and multiplexing barcodes added by the Index PCR in 25 μL containing 12.5 μL of MyTaq Red Mix (Bioline), 0.5 μL of each primer (10 μM), 0.5 μL of DNA template and 11 μL of MBQW at (1) 95 °C (3 min); (2) 8 cycles of 98 °C (20 s), 57 °C (15 s), 72 °C (15 s); (3) 72 °C (5 min). Final amplicons were quantified (Qubit 2.0 Fluorometer (Invitrogen)) and normalised to 4 nM (SequalPrep Normalization Plate Kit, Invitrogen), combined and sequenced (Illumina MiSeq, 300 base paired-ends).

## Sequence processing pipeline and phylogenetic analysis

Paired-end de-multiplexed files were imported into QIIME2 (v.2021.11) on the Apocrita HPC facility at Queen Mary University of London[73] (Supplementary Fig. 9) and processed using DADA2 to trim primers, remove low-quality sequences and chimeras[74]. Sequences were clustered into species-level OTUs at 95% similarity[72], singletons and sequences >356 bp or <333 bp and low-abundance OTUs (<20 reads and in 3 samples or less) were removed. Amino acid sequences were aligned to known $nifH$ and non-$nifH$ references and a phylogenetic tree constructed using COBALT[75]. The primers IGK3/DVV can amplify non-$nifH$ homologues including the chlorophyll synthesis genes $BChL$ and $ChlL$ and these were identified after translating the OTU sequences using "Translate" in MEGA (version 10.2.2). Translation initiation site adjustment and frameshifts were detected using blastp[76]. Amino acid sequences were aligned to known $nifH$ and non-$nifH$ references and a phylogenetic tree constructed using COBALT[75]. The non-$nifH$ OTUs were identified using distinct non-conservative short sequence motifs and visualised using the iTOL tool[77] that appeared as two separate clusters on the phylogenetic tree (see Supplementary Fig. 10). Approximately 82% of the sequences were non-$nifH$ homologues, which is common when using general $nifH$- primers[71]. Non-$nifH$ sequences were removed and q-PCR estimates of $nifH$ gene abundances were corrected for the proportion of non-$nifH$ sequences in each sample.

## Statistical analysis

Statistical analysis and plotting were performed in R[78] using RStudio (Version 1.3.1093). We used generalised additive mixed effects models (GAMMs)[79] to characterise the seasonal patterns in $N_2$ and $N_2O$ saturation, fitting sampling month as a fixed effect and each replicate pond as random effects. We included an interaction term for sampling month by gas ($N_2$ or $N_2O$) to explore any distinct seasonality in $N_2$ and $N_2O$ saturation. Models were ranked by the small sample-size corrected Akaike Information Criterion (AICc) using the 'MuMIn' package (Supplementary Table 1).

Rate data for total $^{15}N_2O$ reduction and $^{15}N$ assimilation were skewed, potentially due to normalising to unit dry biomass which may not account for the true abundance of $N_2$ and $N_2O$ fixers. Therefore, we present 95% of data (2.5% to 97.5% percentiles) for both datasets (Figs. 2a, c, 3a and 4a) and fitted quantile regression models ('quantreg'[80]) rather than mean regression models to the full dataset for rate of $^{15}N$ assimilation to minimise any bias from outliers (median regression lines, Fig. 4a). The difference between the temperature response of $^{15}N_2$ and $^{15}N_2O$ was compared using the 'emmeans' package.

*nifH* Shannon diversity was calculated using the 'estimate_richness' function in the 'phyloseq' package[81] and any changes in the *nifH* community calculated using the 'adonis' function from the 'Vegan' package[82] (with Original UniFrac distance). Principal Coordinates Analysis (PCoA) was used to test the significance of either incubation day or excess $N_2O$ on *nifH* community composition by Permutational multivariate analysis of variance (PERMANOVA) and redundancy analysis (RDA) to ordinate $N_2O$ reduction and *nifH* relative abundance.

## Reporting summary

Further information on research design is available in the Nature Portfolio Reporting Summary linked to this article.

## Data availability

Data generated in this study are provided in the Source Data file. Source data are provided with this paper. The DNA sequences are in the National Center for Biotechnology Information database, under BioProject ID PRJNA984972. Source data are provided with this paper.

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

## Acknowledgements

This study was supported through a PhD Studentship from Queen Mary University of London and additionally by the Leverhulme Trust (RPG-2019-008) to M.T. We thank M. Rouen for designing and installing the warming and data-logging system for the ponds, W. Beaumont for providing the on-site wind speed data, and J. Pretty for routine maintenance of the ponds.

## Author contributions

M.T. and Y.S. conceived the study. Y.S. performed the saturation and $^{15}$N-tracer incubations, analysed the data and wrote the manuscript. Y.Z. performed the 25-day incubation and, with K.J.P., the *nifH* community analysis. I.S. and D.K. provided field and technical support. All authors contributed to revisions of the manuscript.

## Competing interests

The authors declare no competing interests.
