## [Peer Review File · Nature Communications]

Direct biological fixation provides a freshwater sink for N₂OReviewer #1 (Remarks to the Author):

The manuscript titled "Direct biological fixation provides a freshwater sink for N₂O" by Si et al. investigated a novel microbial sink for the potent greenhouse gas nitrous oxide (N₂O): namely, fixation of gaseous N₂O into bioavailable nitrogen.

Using ¹⁵N tracer experiments with biomass from experimental ponds, the authors identified a potentially direct pathway converting N₂O to biomass and dissolved inorganic nitrogen. The authors then analyzed the community containing the N₂ fixation marker gene *nifH* to identify candidate organisms that may be performing this N₂O fixation.

Satisfyingly, the authors found that the rates of N₂O fixation peaked in the wintertime when there was the greatest N₂O undersaturation in the experimental ponds. They also found that N₂O fixation rates dropped off as dissolved inorganic nitrogen increased, and that N₂O fixation is controlled kinetically by the availability of substrate N₂O, as opposed to temperature, allowing for a higher ratio of N₂O fixation to N₂ fixation in the cold. This is another satisfyingly intuitive result: the dissociation energy of the N-N bond in N₂O is half that of the N-N bond in N₂, so it makes sense that organisms fixing N₂O in the cold might have an ecological advantage over their N₂-fixing counterparts. Finally, the authors find that N₂O fixation can occur in an abundance of N₂, so it isn't restricted merely to systems where N₂ isn't available.

General appraisal

So far, the only established microbial sink of N₂O has been N₂O reduction to N₂ via denitrification. Since denitrification is considered an anaerobic process, it does not explain the observed undersaturation of N₂O in many aerobic freshwater and marine environments. This study presents an exciting new alternative (N₂O fixation) which could explain this undersaturation in aerobic waters.

This novel process supported only a tiny fraction of the gross primary productivity in the authors' experiments, so it is unlikely to be relevant to global carbon cycling. It does, however, represent a large proportion (often 100%) of the N₂O consumption in their experiments, and thus is likely to become an important consideration in estimating the global N₂O budget. Furthermore, since N₂O undersaturation and N₂O fixation are stronger in the cold, there is a potential negative feedback with warming. This is significant because N₂O is an extremely potent greenhouse gas and the most important ozone depletion agent of the 21st century.

The noteworthy results of this study are in the ¹⁵N tracer experiments showing the transfer of ¹⁵N from N₂O to biomass, presumably directly (although I have questions about some of the calculations, see below). Supporting these results, the *nifH* analysis comes close to providing candidate organisms for an N₂O-fixing process.

The other noteworthy result, although it receives very little discussion, is the presence of N₂O reduction to N₂ in the authors' presumably aerobic incubations. I do not doubt that it is occurring, but why and how could this anaerobic process be occurring in fully oxygenated experiments? If I misread and the incubations were actually anoxic, this should definitely be clarified in the text. This intriguing result is in agreement with a study showing the presence of *nosZ* in aerobic surface waters of the tropical Pacific, and furthermore that the denitrifying community is capable of switching on N₂O reduction as soon as oxygen concentrations decline (Sun et al. 2021).

In addition, the researchers did an astounding number of incubations (685!) which represent hundreds of hours of work. Because of the vast number of incubations, the authors were able to analyze their results with a level of robustness and nuance only afforded by such a large dataset.

My first major concern about the validity of the results is that half of the authors' experiments contained N₂O reduction to N₂ via denitrification — and, given the high N₂ fixation rates the authors also measured, no doubt some of this N₂ was subsequently converted to biomass. The authors were able to account for this indirect process by pairing ¹⁵N-labelled N₂O experiments with ¹⁵N-labelled N₂ experiments. Their calculation, however, is summarized so briefly that I was

unable to reproduce their numbers (see specific comments, below). If the authors were to lay out this calculation more clearly, they would resolve this concern for me.

My second concern is that by the authors' own admission, there is no known freshwater organism capable of fixing N₂O. The authors present some candidate operational taxonomic units (OTUs) for this process by showing that the abundance of these OTUs correlated with the rates of N₂O consumption, but I worry that we've strayed into the land of post hoc ergo propter hoc. The nifH analysis is helpful, but it by no means fills this important gap. If the authors were to evaluate more thoroughly any confounding factors for the correlations they observe, that would help remedy this concern.

My final concern is a more philosophical one: given that there is so much N₂ in the environment, why would an organism fix N₂O instead? Does the difference in dissociation energies for the N-N bonds in N₂O vs. N₂ really outweigh the scant nature of N₂O in the environment (10 nM N₂O vs. 487 μM N₂)? I would be happy to see the authors address this question more thoroughly in the discussion.

My concerns about the methodology center primarily on the N₂O isotopocule measurements. Measuring ⁴⁵N₂O and ⁴⁶N₂O with isotope ratio mass spectrometry is non-trivial and requires numerous corrections. Please describe more thoroughly which reference materials were used, any corrections that were applied, and the analytical uncertainty and detection limits of your instrument or cite literature that describes your method. Also, it was unclear throughout the manuscript which incubations were 24-hour incubations and which were 25-day incubations.

I am also concerned about isotope dilution of the ¹⁵N-labeled N₂O, which could result in the N₂O fixation rates being underestimated. There is almost certainly N₂O production occurring in these experiments, although it may be limited initially by the small DIN pool. The authors should mention this in the text or provide a calculation showing that it is insignificant.

The major barriers to reproducibility are the inadequate descriptions of the N₂O isotopocule measurements and some of the calculations in the text. Additionally, the authors have not placed their data or code on any repositories, which is becoming standard in the field (and is a requirement for AGU, if not Nature, journals).

The paper is well-written and easy to read. There are a few places where the logic or calculations could be more clearly laid out, which I detail below. I also feel that something is lost by focusing only on the N₂O fixation story, instead of both N₂O fixation and N₂O reduction via denitrification, but I understand that this was a choice in order to fit the paper into the short format for this journal.

Finally, the qPCR and phylogenetic analyses in this study were outside of my area of expertise, so my assessment of these sections of the paper is limited.

Specific comments

Line 33: *273 now, I believe - see IPCC AR6 (Smith et al. 2021)

Line 38: "Ignored" feels like a strong statement. There is a body of research out there on the widespread nature of NosZ. See Sun et al. (2021) and others.

Lines 54-55: Could you list what these analytical artifacts are, and why you believe they are insufficient to explain the observed undersaturation?

Line 82: But there is so much less N₂O than N₂ in the environment.

Line 86: I wonder if it would be better to say N₂O → PON. As it is currently presented, this abbreviation includes ammonification.

Line 90: Same here: N₂O → N₂ → PON

Lines 106-108: Assuming that the recognized N₂ fixing community also mediates N₂O fixation seems like a major assumption. The authors do some clever statistics with their nifH data but in my opinion, the manuscript falls short of fully evaluating this assumption. Why is it reasonable to assume that the N₂ fixing community can also — and does also — fix N₂O?

Line 116: Is this seasonal cycle driven by a seasonal cycle in the saturation concentration (which would be higher in the colder months due to colder temperatures) and constant measured N₂O concentrations, or seasonal variations in the measured N₂O concentrations? Or both?

Lines 117-118: Why is the N₂O saturation minimum at the end of timeseries so much stronger than the minimum during the first winter period?

Line 125: 5±? Figure 1 shows a big range.

Line 125: Cite Redfield (1934)

Line 127: Primary production must be sustained LARGELY by N fixation. 0.85 μM is low but non-zero.

Line 128: This feels like a big leap before you've presented your reasoning for the occurrence of N₂O fixation. Change to, "which MAY have resulted in" or move this later in the text.

Line 129: It seems to me like it could be an artifact of the saturation concentration decreasing with warmer temperatures while the measured [N₂O] remained constant. It sounds like a lot of this reasoning depends on the quality of the [N₂O] measurement. What is your analytical precision for the [N₂O] measurement?

Line 168: Wow that's a lot of incubations!

Line 179: It's really interesting that you observed N₂O reduction in half of your experiments, even though the experiments were all fully oxygenated! Where is this denitrification happening? Are there anoxic microsites occurring within the biomass? I think something is lost by focusing so much on the N₂O fixation story, so expand on this perhaps if you end up resubmitting to a longer-format journal.

Lines 225-234: While I appreciate the effort to show that the assimilation of ¹⁵N from N₂O into biomass exceeded the amount that could have resulted from ¹⁵N₂O → ¹⁵N₂ → ¹⁵N-PON, it is unclear from this section and the data how the authors arrived at the numbers that they report and thus the validity of this threshold remains ambiguous (0.63 μM ¹⁵N₂ based on what? 0.8 nmol N/g/d based on what?). If these calculations were laid out more clearly, I could be convinced that they are really observing N₂O fixation in these experiments. Also, please report the error associated with these estimated thresholds.

Line 227: I appreciate that the authors did kinetics experiments to scale their measured rates to the in situ concentrations of N₂O (which are three orders of magnitude smaller than the ¹⁵N₂O addition). Did the authors vary the ratio of ¹⁵N₂:¹⁵N₂O in their experiments? Also, why "typically"? were there experiments where you made different additions of ¹⁵N₂ and ¹⁵N₂O?

Line 246: insert (507±? nmol N/g/d)

Line 250: 507±?

Line 251: 237±?

Lines 255-256: It's very satisfying that higher rates of N₂O reduction in winter lined up with greater undersaturation in winter.

Line 261: Is the N₂O being fixed as NH₄⁺ or as PON? This is unclear throughout the manuscript.

Line 266: This makes it sound like the controls are shown in Figure 3c - which actually would have been helpful to see.

Line 289: Minor proportion? Don't you have experiments where 100% of the $^{15}\text{N}_2\text{O}$ was reduced to $^{15}\text{N}_2$ via denitrification? (Figure 2b)

Line 298: Does the lack of temperature sensitivity cast any doubt on whether this is a biological process?

Line 327: It seems like a big assumption that the N_2 fixing community is also capable of/responsible for the observed N_2O fixation, when the authors have no freshwater candidates for this process.

Line 388: *half. "Most" is strong for 51%

Line 435: Is "pristine" the limnology term for "oligotrophic"?

Lines 446-449: This is interesting but it feels like an unnecessary aside.

Line 673: How were $^{45}\text{N}_2\text{O}$ and $^{46}\text{N}_2\text{O}$ measured? Were $^{45}\text{N}_2\text{O}$ and $^{46}\text{N}_2\text{O}$ standards used? How were the concentrations $^{45}\text{N}_2\text{O}$ and $^{46}\text{N}_2\text{O}$ converted into total " $^{15}\text{N}-\text{N}_2\text{O}$ "/what does " $^{15}\text{N}-\text{N}_2\text{O}$ " represent? $2 \times ^{45}\text{N}_2\text{O} + ^{46}\text{N}_2\text{O}$? What is the detection limit for their instrument? What is the analytical precision? What corrections were applied to the $^{45}\text{N}_2\text{O}$ and $^{46}\text{N}_2\text{O}$ measurements? Peak size correction applied? Scale normalization? Normalized to a common reference injection?

Lines 677-678: "Only a small fraction of subsamples" makes it sound like you only measured a fraction of your samples. I think you mean, "only a small aliquot of the sample headspace".

Line 680: Are these $^{44}\text{N}_2\text{O}$ standards?

Lines 687-688: What corrections were applied to the $^{15}\text{N}_2$ measurement? Peak size correction applied? Scale normalization? Normalized to a common reference injection?

Lines 691-692: "Detection limit for $^{15}\text{N}_2$ production" makes it sound like this number should represent a rate, not a concentration.

Line 702: what parameters?

Lines 703-704: If you're not going to describe this isotope correction, add a citation to the paper that describes this correction.

Line 705: The mass spectrometer doesn't measure $\delta^{15}\text{N}$; it measures ion current ratios. Describe how you went from raw isotope ratios measured by the mass spec to $\delta^{15}\text{N}$ values, including any corrections for peak area, drift, blanks, etc. (or cite a paper that describes the corrections you applied).

Lines 715-718: This should be an equation. Does this calculation assume that the fraction of ^{15}N in N_2O and ^{15}N in N_2 remains constant over the course of the experiment? If you have biomass and remineralization in your incubations, you likely have some nitrification, which would dilute the $^{15}\text{N}_2\text{O}$ signal and cause your N_2O assimilation rates to be an underestimate.

Lines 735-737: Can you add a supplementary figure showing this interference? This would be extremely useful to others making this measurement.

Line 775: I don't think it would change the interpretation of the results, but it's worth noting that NO interferes with the optode oxygen measurements (Kraft et al. 2022).

Lines 786-787: I thought formaldehyde interfered with the ammonium measurement?

Line 854: GAMM appears earlier in the manuscript - define it there.

Lines 864-865: Since it adds so much variability, why not just present the rates without being normalized to a unit of dry biomass?

Technical corrections

Line 101: Choose either "We show... we hypothesize..." or "We showed... we hypothesized..."

Line 117: Define GAMM

Line 121: Define abbreviations

Line 170: Should this reference be Figure 2c?

Line 171: Figure 2c?

Line 181: typo: Theoretical

Line 294: "experimentally" is redundant here

Lines 467-468: Clumsy sentence. Consider rephrasing.

Line 688: typo: should be $^{15}\text{N}^{14}\text{N}$

Line 777: typo: after

Figures and tables

Figure 1b: Since so much of this paper hinges on these N_2O saturation measurements, I would really like to see error bars on these dots. I realize that it's non-trivial to propagate error through the K_o calculation but at very least I would like to see the errors associated with the $[\text{N}_2\text{O}]$ measurement (assuming that these are much greater than any errors due to uncertainties in the K_o calculation).

Figure 1b: Consider plotting $[\text{N}_2\text{O}]$ and the saturation concentration so we can see what drives this seasonal cycle.

Figure 1c: I understand why you wanted to plot N_2 saturation and N_2O saturation on the same y-axis ranges, but this makes it look like there's no seasonal cycle in N_2 , which is contradictory to what you state above.

Figure 2a: Are these the 25th to 75th percentiles of this subset of data (excluding outliers) or of the full dataset?

Figure 2c: What do these dots represent? The mean across the full dataset?

Figure 2c: The calculation for the dashed red line needs to be more clearly laid out. Also, I would expect there to be an error associated with this value — maybe display this as a shaded area on either side of the line.

Figure 2 caption: Is it misleading to only report the median, given that you also have experiments where there's a lot of N_2O reduction via denitrification? Report the mean, as well.

Table 1: Could you also calculate the predicted maximum ^{15}N -labelling of the N_2O pool resulting from $^{15}\text{N}_2$ fixation, ammonification, and then N_2O production from ^{15}N -labelled NH_4^+ ?

Figure 3c: It would be instructive to include the production of N_2 in this plot.

Figure 3d: I know this is a simplified diagram, but since you have N₂O reduction to N₂ via denitrification in many/half of your experiments, should this diagram also contain nitrate reduction to nitrite and N₂O via denitrification?

Supplementary figure 3: What kind of linear regression is the line in a?

References

Kraft, B., N. Jehmlich, M. Larsen, L. A. Bristow, M. Könneke, B. Thamdrup, and D. E. Canfield. 2022. Oxygen and nitrogen production by an ammonia-oxidizing archaeon. *Science*. doi:10.1126/science.abe6733

Redfield, A. C. 1934. *On the Proportions of Organic Derivatives in Sea Water and Their Relation to the Composition of Plankton*, University Press of Liverpool.

Smith, C., Z. R. J. Nicholls, K. Armour, and others. 2021. The Earth's Energy Budget, Climate Feedbacks, and Climate Sensitivity Supplementary Material, In V. Masson-Delmotte, P. Zhai, A. Pirani, et al. [eds.], *Climate Change 2021: The Physical Science Basis. Contribution of Working Group I to the Sixth Assessment Report of the Intergovernmental Panel on Climate Change*. Cambridge University Press.

Sun, X., A. Jayakumar, J. C. Tracey, E. Wallace, C. L. Kelly, K. L. Casciotti, and B. B. Ward. 2021. Microbial N₂O consumption in and above marine N₂O production hotspots. *ISME J.* 15: 1434–1444. doi:10.1038/s41396-020-00861-2

Reviewer #2 (Remarks to the Author):

I find the research topic of this manuscript very significant and provides really interesting results on the capacity of N₂O fixation by freshwater communities in N-limited experimental ponds. I also find very interesting the comparison of N₂O fixation with canonical N₂ fixation and their dependencies on seasonality and temperature, as well as the search for potential candidates in the diazotrophic community as N₂O sinks. In addition, the development of the experimental designs was very well elaborated and thought out.

My main concern is that although in terms of structure the article is well written, it is tremendously long and many parts of the text in the results and methods sections can be shortened. Articles should be written in a concise manner showing only the main results. Therefore, I strongly suggest reducing both sections without compromising critical information and shuttle the redundant or unnecessary information to the supplementary section. For example, there are sentences in the results that can be eliminated because they are over-understood by looking at the figures or because they are repetitive with the information in the figure captions (in fact, I suggest reducing the figure captions as well). Also, the methodology should not take up more than 5-6 pages of the main document and here there are 16 pages!

On the other hand, it is not clear from the main text when the samples were taken from the ponds for each of the analyses. I deduce from some figures that the moments and/or sampling times were not always the same for each analysis, right? This information must be explicit in the methodology for each analysis. In some parts, the information on sampling times even seems contradictory or confusing. For example, the graphs b and c in Fig. 1 show a sampling period from January to December (what year?), but the title of the figure indicates that the sampling was from November 2019 to April 2022; therefore it does not match. The same happens in Fig. S2 where the graph shows a sampling period for 12 months in a year, but the title of the figure indicates that the sampling was from January 2019 to December 2021 (this means 3 years of sampling!). Please, review carefully this information in all figures. I also suggest including a short paragraph at the beginning of the results contextualizing the experimental design and the sampling to understand the results.

Finally, the discussion seems to me well raised and bounded to the results obtained. Two suggestions:

L388: Delete "in most" (only leave the percentage)

L433-436: N fixation (even high rates) has been demonstrated in N-enriched aquatic systems, so this sentence is not correct. Please, rewrite the sentence.

Detailed replies (in blue) to reviewer comments (in black) on NCOMMS-23-18678A

Reviewer #1 (Remarks to the Author):

The manuscript titled “Direct biological fixation provides a freshwater sink for N₂O” by Si et al. investigated a novel microbial sink for the potent greenhouse gas nitrous oxide (N₂O): namely, fixation of gaseous N₂O into bioavailable nitrogen.

Using ¹⁵N tracer experiments with biomass from experimental ponds, the authors identified a potentially direct pathway converting N₂O to biomass and dissolved inorganic nitrogen. The authors then analyzed the community containing the N₂ fixation marker gene *nifH* to identify candidate organisms that may be performing this N₂O fixation.

Satisfyingly, the authors found that the rates of N₂O fixation peaked in the wintertime when there was the greatest N₂O undersaturation in the experimental ponds. They also found that N₂O fixation rates dropped off as dissolved inorganic nitrogen increased, and that N₂O fixation is controlled kinetically by the availability of substrate N₂O, as opposed to temperature, allowing for a higher ratio of N₂O fixation to N₂ fixation in the cold. This is another satisfyingly intuitive result: the dissociation energy of the N-N bond in N₂O is half that of the N-N bond in N₂, so it makes sense that organisms fixing N₂O in the cold might have an ecological advantage over their N₂-fixing counterparts. Finally, the authors find that N₂O fixation can occur in an abundance of N₂, so it isn't restricted merely to systems where N₂ isn't available.

General appraisal

So far, the only established microbial sink of N₂O has been N₂O reduction to N₂ via denitrification. Since denitrification is considered an anaerobic process, it does not explain the observed undersaturation of N₂O in many aerobic freshwater and marine environments. This study presents an exciting new alternative (N₂O fixation) which could explain this undersaturation in aerobic waters.

This novel process supported only a tiny fraction of the gross primary productivity in the authors' experiments, so it is unlikely to be relevant to global carbon cycling. It does, however, represent a large proportion (often 100%) of the N₂O consumption in their experiments, and thus is likely to become an important consideration in estimating the global N₂O budget. Furthermore, since N₂O undersaturation and N₂O fixation are stronger in the cold, there is a potential negative feedback with warming. This is significant because N₂O is an extremely potent greenhouse gas and the most important ozone depletion agent of the 21st century.

The noteworthy results of this study are in the ¹⁵N tracer experiments showing the transfer of ¹⁵N from N₂O to biomass, presumably directly (although I have questions about some of the calculations, see below). Supporting these results, the *nifH* analysis comes close to providing candidate organisms for an N₂O-fixing process.

The other noteworthy result, although it receives very little discussion, is the presence of N₂O reduction to N₂ in the authors' presumably aerobic incubations. I do not doubt that it is occurring, but why and how could this anaerobic process be occurring in fully oxygenated experiments? If I misread and the incubations were actually anoxic, this should definitely be

clarified in the text. This intriguing result is in agreement with a study showing the presence of *nosZ* in aerobic surface waters of the tropical Pacific, and furthermore that the denitrifying community is capable of switching on N₂O reduction as soon as oxygen concentrations decline (Sun et al. 2021).

First, the reviewer has not misread anything; the incubations were not anoxic but we have added “oxygen-saturated artificial pond water” to lines 555 and 557 for clarification. Second, we had overly simplified the presentation of our denitrification data (i.e., ¹⁵N₂ production from ¹⁵N₂O) to keep the focus on N₂O fixation but have now revised this aspect of our paper. We have been back through all of the data for ¹⁵N₂ production from ¹⁵N₂O, re-evaluated the limit of detection for ³⁰N₂ – showing all the mass spec data – and can now show quite clearly (this is returned to on line 179 below but we supply the full details in reply to the comment for line 289) that we only have significant production of ¹⁵N₂ in the benthic biomass incubations but not the floating and of that, only 30% of the benthic incubations produced measurable ¹⁵N₂ and not the original 51% from all incubations. A denitrification potential in the benthic biomass makes sense as we know oxygen only penetrates <1 cm into the pond sediments *in situ* i.e., the sediments are reactive (Zhu et al., 2020) and that our 12h/12h light/dark incubation-cycle generated oxygen minima overnight that likely facilitated the reduction of N₂O to N₂ via denitrification.

N₂O reduction has been reported in oxic surface waters in the Atlantic (Rees et al., 2021), in relation to Clade II *nosZ* genes, and the oxic-to-anoxic transitioning waters in the Eastern Tropical North Pacific to *nosZ* genes in general (Sun et al., 2021). We have now done qPCR for Clade II *nosZ* genes (primers nosZ1F and nosZ1R) (Henry et al., 2006) and while we got a positive PCR product the counts for Clade II were either very low (three orders of magnitude less than for *nifH*) or unquantifiable in 80 % of our samples. However, if *nosZ* were responsible for the reduction of N₂O then we would expect N₂O assimilation to go through N₂ first which would still leave our measured N₂O assimilation far above the threshold for indirect N₂O fixation – this section has now been fully revised along with Table 1 in accordance with the comments below “My first major concern” and in detail to the query related to lines 225-234.

In addition, the researchers did an astounding number of incubations (685!) which represent hundreds of hours of work. Because of the vast number of incubations, the authors were able to analyze their results with a level of robustness and nuance only afforded by such a large dataset.

Thanks, and yes it was a considerable amount of work.

My first major concern about the validity of the results is that half of the authors’ experiments contained N₂O reduction to N₂ via denitrification — and, given the high N₂ fixation rates the authors also measured, no doubt some of this N₂ was subsequently converted to biomass. The authors were able to account for this indirect process by pairing ¹⁵N-labelled N₂O experiments with ¹⁵N-labelled N₂ experiments. Their calculation, however, is summarized so briefly that I was unable to reproduce their numbers (see specific comments, below). If the authors were to lay out this calculation more clearly, they would resolve this concern for me.

First, as above, we have re-evaluated the limit of detection for ¹⁵N₂ (see Specific Comment for line 289, below) and only have significant production in 30% of benthic incubations. Overall, ¹⁵N₂ content in the controls was not different to that in the ¹⁵N₂O-amended

incubations with floating biomass but significant $^{15}\text{N}_2$ production was detected in incubations with benthic biomass (see Fig. 3 below, revised Fig. 2c and new lines 177 – 183 in the manuscript).

We now also explain the calculation for the threshold of indirect N_2O fixation more clearly and have added the error related to the threshold in reply to your specific comment below and in the revised manuscript, see new lines 184 – 193, revised Fig. 2 and Table 1.

My second concern is that by the authors' own admission, there is no known freshwater organism capable of fixing N_2O . The authors present some candidate operational taxonomic units (OTUs) for this process by showing that the abundance of these OTUs correlated with the rates of N_2O consumption, but I worry that we've strayed into the land of post hoc ergo propter hoc. The *nifH* analysis is helpful, but it by no means fills this important gap. If the authors were to evaluate more thoroughly any confounding factors for the correlations they observe, that would help remedy this concern.

We are aware that our relationship between candidate *nifH* OTUs and N_2O consumption is correlational and does not link cause to effect by definitively identifying the organism(s) that drive N_2O fixation in our ponds – we stated in our cover letter (admittedly not seen by the reviewer) that we had only taken tentative steps towards identifying candidates and in the abstract “only a subset is potentially capable” and are cautious throughout. However, we would point the reviewer to the history of discovering organisms that perform novel processes in the environment, with anammox as the perfect example. From Broda's theoretical proposal that ammonia could be oxidised anaerobically using either nitrate or nitrite (nitrite as it turns out) (Broda, 1977) to the discovery that the organisms that do this were Planktomyces took over 20 years from (Mulder et al., 1995) to (Strous et al., 1999), but that didn't hinder progress with the reaction in the environment (Thamdrup and Dalsgaard, 2002; Trimmer et al., 2003). Anammox bacteria were identified in a similar way to ours, using the developing molecular approaches at the time to determine what groups are most closely associated with a process i.e., correlated and then testing that to see if members of that community can be enriched and isolated from the environment – as we attempted to do here. There is essentially no other way to “find” the organisms that drive “unknown” processes – we need to at least first identify candidate taxa e.g. *Pegethrix* spp as we have done here. Our highlighted organisms are not, at this stage, definitive N_2O fixers and potential confounding factors include the probability that some of the highlighted groups may exploit intermediate or end products of the process or exploit any changes in the ecosystem that the process produces. While we are not convinced that explaining the recognised shortcomings of correlation in more detail – as it is no less speculative – would add anything to a reader's understanding of the work we present here but we do reiterate our attempt to link cause to effect and contrast that to our final correlational identification of potential N_2O candidates and also point out that further work is required in natural waters undersaturated in N_2O . See newly revised lines 440 – 460.

My final concern is a more philosophical one: given that there is so much N_2 in the environment, why would an organism fix N_2O instead? Does the difference in dissociation energies for the N-N bonds in N_2O vs. N_2 really outweigh the scant nature of N_2O in the environment (10 nM N_2O vs. 487 μM N_2)? I would be happy to see the authors address this question more thoroughly in the discussion.

First we would like to reiterate what we have measured in the ponds: maximal undersaturation in N₂O in the cold (see new figure below countering claims of a potential artefact) that is distinct from maximal undersaturation in N₂ in the warm. Ecologically, therefore, N₂O undersaturation appears related to lower energy times of the year be it either in terms of light-limited photosynthesis or low-methane (Zhu et al., 2020) (and/or potentially low-sulfide) concentrations limiting chemosynthesis. In contrast, maximum N₂ fixation occurs in spring and summer when photosynthesis is bountiful. We have now reframed how we present the energetic advantage of fixing N₂O compared to N₂ to show the advantage directly as an energy saving (see new lines 79 – 88):

Here, our ΔG s are defined for freshwater at pH 7 and 10°C for O₂, N₂, N₂O with NO₃⁻ and NH₃ at 10 μ M and 1 μ M, respectively (and see revised SI for full Gibbs energy details). While this ~18% energy saving for fixing N₂O over N₂ may not appear that big, it is worth noting that the original argument for NO₃⁻ being assimilated over N₂ in the ocean i.e., (Falkowski, 1983) and see review by (Knapp, 2012) is, similarly, based on an energy saving of only 21%. Hence, the reviewer could also ask why - in the sunlit, warm, surface-layers of the ocean - haven't all algae evolved *nifH* to exploit the 494 μ M N₂ compared to trace concentrations of NO₃⁻, rather than residing at the base of the mixed layer where light is comparatively limited? There must be an ecological benefit with a 18%-20% saving. Note, Falkowski presented ΔG without defining the conditions, whereas here, in 3 and 4 (below), that energy saving is for the conditions defined above.

Given that shorter days coupled to far lower light intensity reduce the UK (and other temperate northern latitudes) winter light regime to some 15% of that in summer, an 18% saving from fixing N from N₂O vs N₂ could represent a valuable over wintering strategy – *for a subset of the community*. The same energy argument can be made more quantitatively in relation to methanotrophy i.e., some of our *nifH* candidates are *Methylomonas* spp. that are known to fix N₂ (Nguyen et al., 2017).

[ΔG as above but with 1 μ M CH₄] Note, the total energy available from oxidising CH₄ completely to CO₂ is -763 kJ but some 32% is expended fixing C from CH₄ into biomass for growth (Trimmer et al., 2015), leaving -522 kJ. Fixing 1 mol NH₃ from N₂ would take 56% of the remaining (291/522), compared to 47% for N₂O (247/522). Similar energy budgets also apply to sulfur oxidizing bacteria that have recently been implicated in *nifH* expression (Rolando et al., 2023). Accepting the broad principle that all microbes in the environment are starving then, while an entire ecosystem's production could not be sustained by fixing N₂O, some minor, slower-growing organisms could be sustained, particularly at low-light, low-substrate times of the year. We have expanded the discussion to include some of these ideas (see new lines 416 – 426) and included full details in the Supplementary Text 1.

My concerns about the methodology center primarily on the N₂O isotopocule measurements. Measuring ⁴⁵N₂O and ⁴⁶N₂O with isotope ratio mass spectrometry is non-trivial and requires

numerous corrections. Please describe more thoroughly which reference materials were used, any corrections that were applied, and the analytical uncertainty and detection limits of your instrument or cite literature that describes your method. Also, it was unclear throughout the manuscript which incubations were 24-hour incubations and which were 25-day incubations.

We had to look up the umbrella term isotopocule which is exclusively used by those working with N₂O at natural abundance and who are particularly interested in using site preference as a way of identifying/correlating sources and sinks for N₂O – typically in the ocean. Yes, measuring ⁴⁵N₂O and ⁴⁶N₂O at natural abundance is a highly skilled process performed by world leading laboratories – but that is different to the ¹⁵N-enriched incubations used here to trace ¹⁵N into biomass. Yes, we need to be competent – see calibrations below – but with 9 μM 98 atom % ¹⁵N₂O – that is practically all ⁴⁶N₂O – the analysis is not as demanding. That said, perhaps we had relied too heavily on the cited former publications from Trimmer's laboratory and can now see where missing key details for N₂O may have caused confusion. We have added more details on how we measured ⁴⁴N₂O, ⁴⁵N₂O, and ⁴⁶N₂O in reply to your question below, see Fig. 4 and Fig. 5 below, and in the extensively revised methods section of the manuscript, see new lines 584 – 595.

We have also edited the text and changed the heading in the methods for the 25-day incubation to distinguished it from the routine 24-hour incubations more clearly, see new lines 541, 613, 619, and 642.

I am also concerned about isotope dilution of the ¹⁵N-labeled N₂O, which could result in the N₂O fixation rates being underestimates. There is almost certainly N₂O production occurring in these experiments, although it may be limited initially by the small DIN pool. The authors should mention this in the text or provide a calculation showing that it is insignificant.

As above, we think it is important to bear in mind here that we are not working at natural abundance. The added ¹⁵N-N₂O concentration at 9 μM was very high compared to ambient ~12 nM. In addition, the ¹⁵N-labelling of the added ¹⁵N-N₂O was also high (> 98 atom % ¹⁵N) and remained constant during the incubations (i.e., F_{N₂O} = 0.98), which also indicates that ¹⁴N-N₂O production from DIN was insignificant. We return to this below for the comment on lines 715-718 and in relation to Table 1. We can estimate mineralisation and nitrification but, even at a yield of 0.5% N₂O for nitrification, N₂O production would be trivial at 0.5nM d⁻¹ compared to 9μM ¹⁵N-N₂O and isotope dilution is not a concern.

Further, for her complete thesis Yueyue measured the nitrification and denitrification potentials (there is no anammox) in oxic and anoxic incubations, respectively, with pond biomass enriched with DIN. Even here with 10μM ¹⁵NO₃⁻ or ¹⁵NH₄⁺ N₂O production was trivial.

The major barriers to reproducibility are the inadequate descriptions of the N₂O isotopocule measurements and some of the calculations in the text. Additionally, the authors have not placed their data or code on any repositories, which is becoming standard in the field (and is a requirement for AGU, if not Nature, journals).

As mentioned above in “My concerns about the methodology center primarily on the N₂O isotopocule measurements.”, we have added more details on how we measured ⁴⁴N₂O, ⁴⁵N₂O, and ⁴⁶N₂O (see Fig. 4 and Fig. 5 below, and Supplementary Fig. 13) and in the revised methods section of the manuscript, see new lines 584 – 595.

We have now provided the source data but as the R code is only for familiar statistics and drawing the figures, we really don't feel it is necessary.

The paper is well-written and easy to read. There are a few places where the logic or calculations could be more clearly laid out, which I detail below. I also feel that something is lost by focusing only on the N₂O fixation story, instead of both N₂O fixation and N₂O reduction via denitrification, but I understand that this was a choice in order to fit the paper into the short format for this journal.

As above, the denitrification data have been fully revised (see Specific Comment for line 289, below) and denitrification in the benthic biomass expanded on in the text. See new lines 177 – 183.

Finally, the qPCR and phylogenetic analyses in this study were outside of my area of expertise, so my assessment of these sections of the paper is limited.

See opening remarks in relation to our exploration of *nifH* and the original discovery of anammox. Further, very little microbial ecology research ever gets past correlating processes to either gene abundance and/or expression, with only stable isotope probing e.g. with ¹³CH₄ explicitly linking methane oxidation to active methanotrophs via ¹³C-labelled DNA. Of course, the definitive test of form to function in microbiology is pure culture but that would restrict our knowledge to a tiny fraction of prokaryotes on Earth, besides, some, like anammox, resist being cultured to this day.

Specific comments

Line 33: *273 now, I believe - see IPCC AR6 (Smith et al. 2021)

Revised and reference updated. See new lines 33 – 34.

Line 38: "Ignored" feels like a strong statement. There is a body of research out there on the widespread nature of NosZ. See Sun et al. (2021) and others.

We admit that it was not right to say "ignored". We have revised the sentence and referenced both Sun et al. (2021) and Rees et al. (2021) and return to this at both the beginning and end of the discussion. See new lines 38, 372 – 373, and 480 – 481.

Lines 54-55: Could you list what these analytical artifacts are, and why you believe they are insufficient to explain the observed undersaturation?

It is not us proposing the artifacts, rather the two papers we cited literally refer to their measured undersaturation in N₂O as artifacts. (Butler et al., 1989) refers to the measured undersaturation in N₂O in the surface of the South Pacific central gyre as artifacts potentially generated by seasonal cooling, then put a few sentences saying however, seasonal cooling did not generate N₂O undersaturation in the North Pacific central gyre – so it is at best vague. In (Cline et al., 1987), they measured strong undersaturation in N₂O in 6 sites (62% to 86%) and oversaturation in 2 sites. They simply refer to these undersaturation values as artifacts as "no known chemical or biochemical removal mechanisms" can explain such undersaturation in

the surface layers, without any further discussion. Whereas our work and that of Sun et al. (2021) – while contrasting – now offer biological explanations.

Line 82: But there is so much less N₂O than N₂ in the environment.

This part of the text has been revised in reply to the reviewer's general comment above and to now show the energy saving from fixing N₂O versus N₂ directly and which we now expand on in the discussion but do not want to open it up too much in the introduction. See new lines 79 – 88, and 416 – 426.

Line 86: I wonder if it would be better to say N₂O → PON. As it is currently presented, this abbreviation includes ammonification.

We set out our rationale for N₂O fixation potentially being related to N₂ fixation, including nitrogenase, in the introduction (lines 67 to 88, including equations 1 and 2 which have now been revised). If N₂O fixation follows canonical N₂ fixation then - *at the point it is fixed in the cell* - the product will be NH₃/NH₄⁺. This is distinct from ammonification i.e., remineralisation of organic N to NH₄⁺ and we think people would recognize fixation as being distinct from ammonification. Some of that NH₄⁺ will in turn be used to synthesise organic N i.e., amines, amino acids, proteins – in total, PON. A fraction of that NH₄⁺ also appears to “leak” to the surrounding water where it is either assimilated by other members of the community (PON) or oxidised to NO₂⁻ and NO₃⁻ which, in turn, could also be assimilated into PON. All of which we are trying to convey in the simple Fig. 3d. We have introduced this more clearly on lines 254 to 259 and reordered the text on lines 254 to 265.

Line 90: Same here: N₂O → N₂ → PON

As above, fixation of N to NH₄⁺ is distinct from ammonification.

Lines 106-108: Assuming that the recognized N₂ fixing community also mediates N₂O fixation seems like a major assumption. The authors do some clever statistics with their *nifH* data but in my opinion, the manuscript falls short of fully evaluating this assumption. Why is it reasonable to assume that the N₂ fixing community can also — and does also — fix N₂O?

Our assumption is not actually that big or completely without foundation. On lines 61 to 88 we set out what little is known about N₂O fixation to date including the work showing ¹⁵N₂O assimilation by two N₂-fixing marine isolates *Trichodesmium* and *Crocospaera* (whereas here we have natural communities). We also mention the 1950's accounts of soybean root nodules i.e., recognised legumes, also assimilating ¹⁵N₂O and how N₂O has been shown to be a competitive inhibitor of the nitrogenase complex i.e., N₂O must share similarities with N₂ from the perspective of nitrogenase. We then state how it is hard to fix N₂ in the cold and how the lower energy required to fix N₂O could provide an advantage over N₂, especially in the cold. All in all, therefore, we do not think it unreasonable to at least begin by assuming N₂O fixation could be mediated by *nifH* – as above for the general comments, where else can we start? We do now frame two hypotheses right at the end of the introduction asking whether N₂O fixation is mediated by the whole *nifH* community i.e., just randomly fixing N₂O and N₂ in proportion to their concentrations or whether N₂O fixation is more likely mediated by a subset and which we return to in the discussion. See new lines 109 – 112 and 450 – 460 in the Introduction and Discussion, respectively.

Line 116: Is this seasonal cycle driven by a seasonal cycle in the saturation concentration (which would be higher in the colder months due to colder temperatures) and constant measured N₂O concentrations, or seasonal variations in the measured N₂O concentrations? Or both?

The measured concentrations are not constant, see reply below to the related more detailed comment on line 129. It is not an artefact.

Lines 117-118: Why is the N₂O saturation minimum at the end of timeseries so much stronger than the minimum during the first winter period?

We apologise for any confusion caused here as we were simply trying to show the data at different, seasonal times (months) of the year. The ponds are in Dorset which is a 2-3h train journey from Queen Mary in London and field work was seriously impacted by the pandemic and lock-down in March 2020. See the table below, sampling month is therefore not continuous and not in order of time, but scattered across November 2019 to April 2022 and has been rearranged by month – regardless of year – simply to plot the seasonal data. The overall effect of seasonal variation in temperature on either gas is still apparent in Fig. 1d and Fig. 1e, regardless of year. The N₂O saturation minimum in December, 2020, therefore likely reflects natural, annual variation to January 2022 i.e., they are not at the beginning and end of a continuous time series. We have updated the legend for Fig. 1 to reflect this. We have also added a simple table (see below) to the Supplementary information (Supplementary Table 2).

Table 1. Sampling dates (year & month) for gas saturations.

Year	Month	Gas measured
2019	11	N ₂ O
2020	8, 9, 12	N ₂ O, N ₂
2021	6, 7, 9, 10	N ₂ O, N ₂
2022	1, 2, 3, 4	N ₂ O, N ₂

Line 125: 5±? Figure 1 shows a big range.

True, as boxplots show the full extent of the data, but the top and bottom of the box give the 25% and 75% quantiles and, overall, 91% of the data were below Redfield.

Line 125: Cite Redfield (1934)

Done

Line 127: Primary production must be sustained LARGELY by N fixation. 0.85 μM is low but non-zero.

Done

Line 128: This feels like a big leap before you've presented your reasoning for the occurrence of N₂O fixation. Change to, "which MAY have resulted in" or move this later in the text.

Done

Line 129: It seems to me like it could be an artifact of the saturation concentration decreasing with warmer temperatures while the measured [N₂O] remained constant. It sounds like a lot of this reasoning depends on the quality of the [N₂O] measurement. What is your analytical precision for the [N₂O] measurement?

What we are reporting is not an artefact. Based on your suggestion, we have now plotted (Fig. 1, below) the measured concentrations (blue dots) and equilibration concentrations (K_0 , black line, based on (Weiss and Price, 1980)) for N_2O against temperature. Although we measured the temperature in each pond at the time of each gas sampling and are confident of little variation, we have applied a ± 1 °C variation in temperature to illustrate possible uncertainties in the equilibration concentration, K_0 (shaded grey area).

We then fitted a simple first-order linear model to the measured N_2O concentrations (blue line, slope = -0.36, $p < 0.001$) which we can see differs to the equilibration N_2O concentrations (black line, 2nd order quadratic) and in the shape of their response to temperature. The measured and equilibration concentrations are more different at colder temperatures, showing that N_2O is more undersaturated in the cold. We now provide the concentration data as Supplementary Fig. 1. The analytical precision for the peak area of the N_2O is $\sim 2\%$ (coefficient of variation, $n = 10$), which we have added to the methods section of the manuscript (see revised lines 517 – 518).

Figure 1 | The measured concentration of dissolved N_2O in the pond water (blue dots) and equilibration concentration of N_2O (black line) as a function of temperature.

Line 168: Wow that's a lot of incubations!

Thanks, better to be sure.

Line 179: It's really interesting that you observed N_2O reduction in half of your experiments, even though the experiments were all fully oxygenated! Where is this denitrification happening? Are there anoxic microsites occurring within the biomass? I think something is lost by focusing so much on the N_2O fixation story, so expand on this perhaps if you end up resubmitting to a longer-format journal.

As above in reply to "The other noteworthy result" and see full details in reply to comment on line 289.

Lines 225-234: While I appreciate the effort to show that the assimilation of ^{15}N from N_2O into biomass exceeded the amount that could have resulted from $^{15}N_2O \rightarrow ^{15}N_2 \rightarrow ^{15}N\text{-PON}$, it

is unclear from this section and the data how the authors arrived at the numbers that they report and thus the validity of this threshold remains ambiguous (0.63 μM $^{15}\text{N}_2$ based on what? 0.8 nmol N/g/d based on what?). If these calculations were laid out more clearly, I could be convinced that they are really observing N_2O fixation in these experiments. Also, please report the error associated with these estimated thresholds.

We have fully revised Table 1 and its legend to make this as clear as possible and also the main text. See below for ease and also see lines 184 – 193 in the main text. Further, we have now added a shaded area in Fig. 2b to show the 95% C.I. associated with the estimation of the threshold, which gives a range from 0.69 to 0.92 nmol $\text{g}^{-1} \text{d}^{-1}$.

Table 1. Rationalising N_2O assimilation as direct N_2O fixation. Ambient background concentrations for $^{14}\text{N}_2$ and $^{14}\text{N}_2\text{O}$ in both our $^{15}\text{N}_2$ and $^{15}\text{N}_2\text{O}$ treatments were $\sim 487 \mu\text{M}$ and $0.01 \mu\text{M}$, respectively. We added both $^{15}\text{N}_2$ and $^{15}\text{N}_2\text{O}$ at $9 \mu\text{M}$ (>98 atom % ^{15}N), resulting in initial ^{15}N labelling of the $^{15}\text{N}_2$ and $^{15}\text{N}_2\text{O}$ pools of 0.018 and 0.98 (F_{N_2} and $F_{\text{N}_2\text{O}}$, respectively). If $^{15}\text{N}_2\text{O}$ assimilation was indirect, and $^{15}\text{N}_2\text{O}$ was first reduced to $^{15}\text{N}_2$, then at most $0.63 \mu\text{M}$ $^{15}\text{N}_2$ would have been produced and $F_{\text{N}_2'}$ would have been ≤ 0.0013 . Accordingly, the absolute upper threshold (in red) for indirect $^{15}\text{N}_2\text{O}$ fixation – in proportion to that directly with $^{15}\text{N}_2$ (F_{N_2}) – would have been 0.8 i.e., $[(0.0013/0.018) \times 11.5] \text{ nmol N g}^{-1} \text{d}^{-1}$, which is far lower than our measured rates for $^{15}\text{N}_2\text{O}$ assimilation (in blue, $5.3 \text{ nmol N g}^{-1} \text{d}^{-1}$, on average, Fig. 2b).

Treatment	Process	Frequency of ^{15}N -labelling	^{15}N assimilation (nmol N $\text{g}^{-1} \text{d}^{-1}$)
Direct F_{N_2} and $F_{\text{N}_2\text{O}}$ or indirect $F_{\text{N}_2'}$			
$^{15}\text{N}_2$	Direct N_2 fixation	$F_{\text{N}_2} = 0.018 = [^{15}9\mu\text{M} / (^{15}9\mu\text{M} + ^{14}487\mu\text{M})]$	11.5
$^{15}\text{N}_2\text{O}$	Direct N_2O fixation	$F_{\text{N}_2\text{O}} = 0.98 = [^{15}9\mu\text{M} / (^{15}9\mu\text{M} + ^{14}0.01\mu\text{M})]$	5.3
$^{15}\text{N}_2\text{O}$	*Indirect N_2O fixation	$F_{\text{N}_2'} = 0.0013 = [^{15}0.63\mu\text{M} / (^{15}0.63 + ^{14}487\mu\text{M})]$	≤ 0.8

*With the predicted maximum ^{15}N -labelling of the N_2 pool ($F_{\text{N}_2'}$) resulting from the maximum reduction of $^{15}\text{N}_2\text{O}$ to $^{15}\text{N}_2$.

Line 227: I appreciate that the authors did kinetics experiments to scale their measured rates to the in situ concentrations of N_2O (which are three orders of magnitude smaller than the $^{15}\text{N}_2\text{O}$ addition). Did the authors vary the ratio of $^{15}\text{N}_2$: $^{15}\text{N}_2\text{O}$ in their experiments? Also, why “typically”? were there experiments where you made different additions of $^{15}\text{N}_2$ and $^{15}\text{N}_2\text{O}$?

Sorry for any ambiguity here. We did not vary the ratio of $^{15}\text{N}_2$: $^{15}\text{N}_2\text{O}$ in our incubations. We have rephrased the text to say “We added both $^{15}\text{N}_2$ and $^{15}\text{N}_2\text{O}$ at $9 \mu\text{M}$...” We have also shortened the table legend overall in line with reviewer 2’s comments. See new line 227 – 232.

Line 246: insert (507 \pm ? Nmol N/g/d)
Done.

Line 250: 507 \pm ?
Done.

Line 251: 237 \pm ?
Done.

Lines 255-256: It's very satisfying that higher rates of N₂O reduction in winter lined up with greater undersaturation in winter.

Ok.

Line 261: Is the N₂O being fixed as NH₄⁺ or as PON? This is unclear throughout the manuscript.

See reply to this point above on lines 86 and 90. We have introduced this more clearly on lines 254 to 259 and reordered the text on lines 254 to 265.

Line 266: This makes it sounds like the controls are shown in Figure 3c – which actually would have been helpful to see.

The production of DIN in the ¹⁵N₂O incubations was calculated by subtracting DIN concentration in the controls from that in ¹⁵N₂O incubations ($Tf_{15N2O} - Tf_{Control}$). For simplicity, we only present the DIN production for ¹⁵N₂O-amended incubations in the figure (Fig. 3b now, after we rearranged the text and figure to make it clearer).

Line 289: Minor proportion? Don't you have experiments where 100% of the ¹⁵N₂O was reduced to ¹⁵N₂ via denitrification? (Figure 2b)

We have taken this opportunity to revisit all of our N₂ data, going back to the original *m/z* 30 signals measured on our Delta V Plus. We have compiled the raw data from all batches of measurements used in this study and show the ratio of *m/z* 30 to total *m/z* ($\Sigma 28+29+30$) over the sequence of measurement (Fig. 2, below). The ratio of 30 *m/z* to total *m/z* is well-known to drift over time which we correct by inserting air standards every 10 samples throughout each measurement sequence (and see line 687, below). Even after drift correction, however, the overall pattern in the ratio does not change (2 panels on the left vs. 2 panels on the right, respectively, below) and we continue with the raw, uncorrected data.

Figure 2 | The ratio of ³⁰N₂ to total N₂ expressed as signal of *m/z* 30 to the total *m/z* ($\Sigma 28+29+30$) in our biomass incubations. A, without drift correction and b, with drift correction for *m/z* 30.

Using the raw data, we plotted the ratio of m/z 30 to total m/z ($\Sigma 28+29+30$) in the control and $^{15}\text{N}_2\text{O}$ treatments (Fig. 3, below). The horizontal line is the mean while the shaded area is the 95% confidence interval from the mixed effect model (see Supplementary Fig. 5). We then used the 95% CI for the controls to define a limit of detection for significant enrichment i.e., $\Delta^{30}\text{N}_2$. Here, our LoD for $\Delta^{30}\text{N}_2$ is $\sim 0.14 \mu\text{M}$ which is comparable to that reported before, e.g., $< 0.1 \mu\text{M}$ (Thamdrup and Dalsgaard, 2000). In our manuscript we had previously put the LoD as $0.03 \mu\text{M}$, which is the LoD for the absolute amount for $^{30}\text{N}_2$ and is not a good measurement for $\Delta^{30}\text{N}_2$ – sorry, this was a breakdown in communication between the PhD (Si) and supervisor (Trimmer). We have now revisited our $^{15}\text{N}_2$ data using the revised LoD (see revised Fig. 2c in the manuscript).

For the floating biomass incubations, the ratio of m/z 30 to total m/z ($\Sigma 28+29+30$) in the $^{15}\text{N}_2\text{O}$ treatment was not statistically different to that in the controls ($P = 0.74$) but we can see statistically significant enrichment in m/z 30 for the $^{15}\text{N}_2\text{O}$ treatment with benthic biomass ($P = 0.007$).

On average, the production of $^{30}\text{N}_2$ from $^{15}\text{N}_2\text{O}$ in the benthic incubations was $0.27 \mu\text{M}$ (Fig. 3b, below), which would result in $0.29 \text{ nmol N g}^{-1} \text{ d}^{-1}$ of ^{15}N assimilation if all of that $^{30}\text{N}_2$ was subsequently fixed. In contrast, this 0.29 nmol N is far lower than the rate of ^{15}N assimilation that we measured ($5.3 \text{ nmol N g}^{-1} \text{ d}^{-1}$) with $^{15}\text{N}_2\text{O}$, which means that even in the benthic biomass incubations that showed significant $^{15}\text{N}_2$ production, $^{15}\text{N}_2\text{O}$ assimilation still appears to have been mainly direct. The production of $^{30}\text{N}_2$ in some (30%) benthic incubations is not surprising given that we would expect benthic biomass/sediment to have a stronger denitrification potential than the floating biomass especially on a 12h/12h light-dark cycle.

For simplicity, in the main text we refer to $^{30}\text{N}_2$ as just $^{15}\text{N}_2$ and have also updated Fig. 2c to show the difference in $^{15}\text{N}_2$ production from $^{15}\text{N}_2\text{O}$ between the floating and benthic biomass in the same normalised units.

Figure 3 | The ratio of $^{30}\text{N}_2$ to total N_2 expressed as raw signal of m/z 30 to the total m/z ($\Sigma 28+29+30$) in all of our biomass incubations.

Line 298: Does the lack of temperature sensitivity cast any doubt on whether this is a

biological process?

While it may appear unusual compared to more familiar biological processes such as respiration that typically demonstrate Q_{10} responses to temperature it is not unprecedented. For example, some chemoautotrophic processes that are highly substrate limited i.e., methanotrophy show a far stronger response to methane than temperature – if at all the latter. Here, any community fixing N_2O is going to be very limited by N_2O in the ponds and even though the incubations were done under elevated N_2O they lack any innate response to temperature. We do not discuss this at this point in the text but do on lines 399 – 403 in the discussion. Such strong substrate limitation for N_2O reduction is clearly demonstrated by our kinetic response from 9 nM to 20,000 nM which we believe is clearly biological. There are no reduced chemicals at high enough concentrations in the water, trace NH_4^+ and perhaps S, to drive abiotic N_2O reduction. Besides which, any abiotic, chemical reduction would not result in ^{15}N being assimilated into PON which is what we have measured.

Line 327: It seems like a big assumption that the N_2 fixing community is also capable of/responsible for the observed N_2O fixation, when the authors have no freshwater candidates for this process.

As above in reply to lines 106 to 108 and the opening general comments in relation to our correlational approach to identifying *nifH* candidates. People rarely have definitive microbes over likely candidates for new processes. As above, see new lines 109 – 112 and 450 – 460 in the Introduction and Discussion, respectively.

Line 388: *half. “Most” is strong for 51%

See several instances above in relation to this point. We have fully revised the denitrification data and this figure has now been revised to significant $^{15}N_2$ production only applying to 30% of the benthic biomass incubations (Fig. 3, above).

Line 435: Is “pristine” the limnology term for “oligotrophic”?

No, the term oligotrophic applies equally to limnology and oceanography it is just that we feel pristine is more readily suited to the broader, generalist audience that we are appealing to here. In light of reviewer 2, however, this passage has now been revised.

Lines 446-449: This is interesting but it feels like an unnecessary aside.

We included this in case our paper got reviewed by Marcel Kuypers who derived those free energies (Kuypers et al., 2018) but it is an aside and has now been removed. The remainder of the material here has now been combined with that in the previous paragraph – see new lines 427 – 439.

Line 673: How were $^{45}N_2O$ and $^{46}N_2O$ measured? Were $^{45}N_2O$ and $^{46}N_2O$ standards used? How were the concentrations $^{45}N_2O$ and $^{46}N_2O$ converted into total “ $^{15}N-N_2O$ ” /what does “ $^{15}N-N_2O$ ” represent? $2 \times ^{45}N_2O + ^{46}N_2O$? What is the detection limit for their instrument? What is the analytical precision? What corrections were applied to the $^{45}N_2O$ and $^{46}N_2O$ measurements? Peak size correction applied? Scale normalization? Normalized to a common reference injection?

Ok, point taken, we could have included a few more details but we did say that $^{15}\text{N}_2\text{O}$ was measured by CF-IRMS and cite (Nicholls et al., 2007) for further details and give a lot more besides. $^{15}\text{N-N}_2\text{O}$ refers to total $^{15}\text{N}_2\text{O}$ i.e., $^{45}\text{N}_2\text{O} + 2 \times ^{46}\text{N}_2\text{O}$ and we used the reduction in $^{15}\text{N-N}_2\text{O}$ to work out the total reduction of N_2O .

The calibration was performed with standards containing known amounts of natural abundance $^{44}\text{N}_2\text{O}$ (Nicholls et al., 2007). We injected different volumes of 96 ppm (BOC, UK) N_2O into air-filled vials as standards, with the amount and peak area of total N_2O ($\Sigma m/z$ 44, 45, 46 i.e., $^{44}\text{N}_2\text{O}$, $^{45}\text{N}_2\text{O}$, and $^{46}\text{N}_2\text{O}$) in the vials then calculated by the excess over air as per (Thamdrup and Dalsgaard, 2000). The relationship between the amount and peak area of total N_2O is linear, and our measurement of the subsamples from the incubations is within the range of the calibration (Fig. 4 below). From previous repeat calibrations, we also know that the relationship is linear at least between 0.1 to 23 nmol of total N_2O . The analytical precision for the peak area of the N_2O measurement is $\sim 1\%$ (coefficient of variation) and the limit of detection for $^{45}\text{N}_2\text{O}$ and $^{46}\text{N}_2\text{O}$ is ~ 0.6 pmol.

We measured the m/z ratios for $^{44}\text{N}_2\text{O}$, $^{45}\text{N}_2\text{O}$, and $^{46}\text{N}_2\text{O}$ in sub-samples of headspace by injecting 10 μL of headspace into air-filled, 12 mL Exetainers. The sample is then cryo-focused in liquid nitrogen using an automated trace gas pre-concentrator (PreCon, Thermo-Finnigan), water trapped and N_2O separated from CO_2 before going through the CF-IRMS (Delta V Plus, Thermo-Finnigan). From the peak area for m/z 44, 45, and 46 of our incubated samples, we can calculate the frequency of ^{15}N in N_2O i.e. $F_{\text{N}_2\text{O}} = 0.981 \pm 0.0004$ (mean \pm s.e.) which matches the ^{15}N -labelling we expected for the purity of $^{15}\text{N}_2\text{O}$ cylinder (> 98 atom %, Cambridge Isotope Laboratories, Inc., USA). In addition, as we also measured the bulk concentration of N_2O in the $^{15}\text{N}_2\text{O}$ treatments and controls, using GC/ μECD , we can also estimate $F_{\text{N}_2\text{O}}$ by $(C_{\text{treatment}} - C_{\text{control}}) \cdot 0.98 / C_{\text{treatment}}$. This also gives us $F_{\text{N}_2\text{O}} = 0.979$, where 0.98 is the purity of $^{15}\text{N-N}_2\text{O}$ (as above). This means that the signal for m/z 44, 45, and 46 measured on the Precon and CF-IRMS is proportional to the amount of 44, 45, and 46. Therefore, using a natural abundance $^{44}\text{N}_2\text{O}$ standard for calibrating $^{45}\text{N}_2\text{O}$ and $^{46}\text{N}_2\text{O}$ is suitable in this application at high atom % enrichment.

Figure 4 | Calibration curve made with known amounts of N₂O showing a linear relationship between the total peak area and nmol of N₂O. The blue dot denotes the average amount of N₂O in the subsamples taken from our incubations for analysis of ⁴⁶N₂O concentration, where the barely visible vertical line within the dot is the standard error.

Finally, while we do not routinely calibrate the mass-spec using high-purity ¹⁵N₂O – as is typical for all work with ¹⁵N-enriched N₂ or N₂O – it is quite clear that the respective sensitivity in total peak area to nmol ¹⁵N₂O (total peak area dominated by *m/z* 46) or nmol ¹⁴N₂O (total peak area dominated by *m/z* 44) is the same (*p* = 0.88) – Fig. 5. We have also put this in the Supplementary Information (Supplementary Fig. 13).

Figure 5 | N₂O measured on our CF-IRMS showing the increase in total peak area to nmol ¹⁵N₂O (total peak area dominated by *m/z* 46) or nmol ¹⁴N₂O (total peak area dominated by *m/z* 44). All vials were pre-flushed with air which resulted in an intercept of 0.38. The lines are simple first order linear regression models with 95% C.I. on the slopes (sensitivity) in brackets.

Lines 677-678: “Only a small fraction of subsamples” makes it sound like you only measured a fraction of your samples. I think you mean, “only a small aliquot of the sample headspace”. Yes, that is what we mean. We can see why the original text is confusing and have edited the text accordingly.

Line 680: Are these ⁴⁴N₂O standards?

Yes, these are natural abundance standards as is typical for all work in this field with ¹⁵N-enriched N₂ or N₂O, see above for details. We have added this to line 591.

Lines 687-688: What corrections were applied to the ¹⁵N₂ measurement? Peak size correction applied? Scale normalization? Normalized to a common reference injection?

The measurement and calculation of $^{15}\text{N}_2$ enrichment in treated samples relative to controls goes back to (Thamdrup and Dalsgaard, 2000) and we effectively adopted the same approach for calculating ^{15}N enrichment in the N_2O , above. Peak size was corrected by the calibration factor (signal/mole N_2) of the CF-IRMS (Delta V Plus, Thermo-Finnigan) based on the concentration of N_2 and the peak area of air standards. Drift correction is also applied by inserting air standards every 10 samples throughout each measurement run. We have added more details to the methods on new lines 598 – 605.

Lines 691-692: “Detection limit for $^{15}\text{N}_2$ production” makes it sound like this number should represent a rate, not a concentration.

We have edited the sentence and now have “Changes in the concentration of $^{15}\text{N}_2$ ($\Delta^{15}\text{N}_2$, nmol N d^{-1}) were calculated by the excess $^{15}\text{N}_2$ in $^{15}\text{N}_2\text{O}$ treatments compared to the controls, where $\Delta^{15}\text{N}_2 = \Delta^{29}\text{N}_2 + 2 \times \Delta^{30}\text{N}_2$. The limit of detection for $\Delta^{15}\text{N}_2$ in the incubations is $\sim 0.14 \mu\text{M}$ ”. See revised lines 602 – 605.

Line 702: what parameters?

We have revised the text, removed the vague term “parameters” and cited (Barneche et al., 2021) for details related to the measurement of ^{15}N assimilation into the biomass. See the reply to “Line 705” below. See revised lines 610.

Lines 703-704: If you’re not going to describe this isotope correction, add a citation to the paper that describes this correction.

See next related point below.

Line 705: The mass spectrometer doesn’t measure $\delta^{15}\text{N}$; it measures ion current ratios. Describe how you went from raw isotope ratios measured by the mass spec to $\delta^{15}\text{N}$ values, including any corrections for peak area, drift, blanks, etc. (or cite a paper that describes the corrections you applied).

Yes, for N_2 and N_2O we use the raw ion currents (as above in reply to line 289) measured on the Thermo Delta V Plus. For routine bulk biomass measurements of both $\delta^{15}\text{N}$ and $\delta^{13}\text{C}$, however, we use a Sercon, Integra2 that is designed explicitly for this purpose. When calibrated, and standards run for drift correction within a run, the Integra2 unit provides the data as actual $\delta^{15}\text{N}$ and $\delta^{13}\text{C}$ values relative to international standards and it is these values that we have used to derive excess ^{15}N atom %. Where excess is the difference between ^{15}N atom % in control biomass samples compared to biomass incubated with either ^{15}N enriched N_2 or N_2O . We also cite Barneche et al 2021 where the conversion of $\delta^{15}\text{N}$ to atom % ^{15}N in Trimmer’s laboratory was explained for the first time. See new lines 610 – 619.

Lines 715-718: This should be an equation. Does this calculation assume that the fraction of ^{15}N in N_2O and ^{15}N in N_2 remains constant over the course of the experiment? If you have biomass and remineralization in your incubations, you likely have some nitrification, which would dilute the $^{15}\text{N}_2\text{O}$ signal and cause your N_2O assimilation rates to be an underestimate.

The description about total assimilation of gases is not needed for the text as we do not refer to it elsewhere. We have now removed this and apologise for any confusion caused by this.

Our oxic incubations were enriched to ~9 μM of ~98 atom % $^{15}\text{N}\text{-N}_2\text{O}$ which is practically all $^{46}\text{N}_2\text{O}$ and any $^{14}\text{N}\text{-N}_2\text{O}$ produced, if at all, from the low concentration of DIN would be minor compared to that $^{15}\text{N}\text{-N}_2\text{O}$ enrichment. In addition, as above, N_2O production from nitrification and denitrification at ambient DIN in our ponds is often undetectable and any nM production would not affect the ^{15}N -labelling i.e. $F_{\text{N}_2\text{O}}$ remained at 0.98 throughout.

Lines 735-737: Can you add a supplementary figure showing this interference? This would be extremely useful to others making this measurement.

In response to reviewer 2's request that we shorten the text, especially for the Methods, we have moved the detail for the formaldehyde interference to Supplementary Information and included a calibration figure (Supplementary Fig. 14) showing the effect.

Line 775: I don't think it would change the interpretation of the results, but it's worth noting that NO interferes with the optode oxygen measurements (Kraft et al. 2022).

We do not believe it would change the interpretation either. Compared to Kraft et al where both O_2 and NO concentrations (see SI Fig. S7) were present in nM (0 to 200 nM), and in which case the correction for NO would be necessary, here, in our oxic incubations, with O_2 between ~180 to 550 μM (light to dark) any interference from nM NO (if at all) would be negligible.

Lines 786-787: I thought formaldehyde interfered with the ammonium measurement?

The Methods were very long and perhaps you missed it – first a sub-sample of water from the serum bottle was transferred to a 3 mL gas-tight vial which was then fixed with formaldehyde for later analysis of N_2O and the remaining water in the serum bottle then filtered and frozen for later analysis of ammonium etc., i.e., no formaldehyde interference.

Line 854: GAMM appears earlier in the manuscript – define it there.

Done

Lines 864-865: Since it adds so much variability, why not just present the rates without being normalized to a unit of dry biomass?

It would be counter intuitive to most readers (and the authors) to talk about assimilation of N into biomass without presenting the data in terms of biomass, yes, ocean water column production and/or consumption of both N_2 and N_2O is often referred to in units of nM h^{-1} etc., but that would not be appropriate here having purposely added biomass.

Technical corrections

Line 101: Choose either “We show... we hypothesize...” or “We showed... we hypothesized...”

Done and thanks for spotting this.

Line 117: Define GAMM

Done.

Line 121: Define abbreviations

Done.

Line 170: Should this reference be Figure 2c?

We have reordered Fig.2 i.e., switched Fig.2b and Fig.2c with our new figure for $^{15}\text{N}_2$ production and revised the text accordingly. See revised line 170 – 206.

Line 171: Figure 2c?

As above for “Line 170”. See revised line 170 – 206.

Line 181: typo: Theoretical

Thanks, corrected.

Line 294: “experimentally” is redundant here

Yes, removed.

Lines 467-468: Clumsy sentence. Consider rephrasing.

Yes, rewritten along with this entire paragraph. See lines 440 – 460.

Line 688: typo: should be $^{15}\text{N}^{14}\text{N}$

Thanks, done.

Line 777: typo: after

Thanks, done.

Figures and tables

Figure 1b: Since so much of this paper hinges on these N_2O saturation measurements, I would really like to see error bars on these dots. I realize that it’s non-trivial to propagate error through the K_o calculation but at very least I would like to see the errors associated with the $[\text{N}_2\text{O}]$ measurement (assuming that these are much greater than any errors due to uncertainties in the K_o calculation).

As above for comments on lines 116 and 129, we now include a plot of both saturation and measured N_2O concentrations as Supplementary Fig. 1 including an estimate of error in K_o for a 1°C variation in temperature. This clearly shows both concentrations changing with temperature over the year and with the greatest difference being between them in the cold. We would rather keep 1b as the simpler product of the two i.e., saturation, as we believe this will be more familiar to a wider audience.

Figure 1b: Consider plotting $[\text{N}_2\text{O}]$ and the saturation concentration so we can see what drives this seasonal cycle.

As above.

Figure 1c: I understand why you wanted to plot N_2 saturation and N_2O saturation on the same y-axis ranges, but this makes it look like there’s no seasonal cycle in N_2 , which is contradictory to what you state above.

We have redrawn panel c to extend the y-axis enough to show the seasonal variation in N_2 saturation and noted the different y-axes scales in the legend. See new Fig. 1.

Figure 2a: Are these the 25th to 75th percentiles of this subset of data (excluding outliers) or of the full dataset?

The boxplots show the 25th to 75th percentiles of 95% of the full dataset (2.5% to 97.5% percentiles).

Figure 2c: What do these dots represent? The mean across the full dataset?

The dots are means \pm s.e. from 95% (2.5% to 97.5% percentiles) of the dataset. We have added “Data plotted are means \pm s.e. from 95% (2.5% to 97.5% percentiles) of the dataset.” To the figure legend.

Figure 2c: The calculation for the dashed red line needs to be more clearly laid out. Also, I would expect there to be an error associated with this value — maybe display this as a shaded area on either side of the line.

As per the detailed comments above, we have added 95% confidence intervals to the threshold in Fig. 2b, rephrased both the title and explanation to Table 1 and rewritten the explanation of the calculation in the main text to make it clearer. See new lines 184 – 193 and 227 – 234.

Figure 2 caption: Is it misleading to only report the median, given that you also have experiments where there’s a lot of N₂O reduction via denitrification? Report the mean, as well.

As above, we have completely revised the ¹⁵N₂ data and how we present them in Fig. 2c. See revised Fig. 2 and lines 177 – 183.

Table 1: Could you also calculate the predicted maximum ¹⁵N-labelling of the N₂O pool resulting from ¹⁵N₂ fixation, ammonification, and then N₂O production from ¹⁵N-labelled NH₄⁺?

First, we can use the TIN production data over the first 3-days of the 25-day incubation to better estimate rates ($\mu\text{M N d}^{-1}$) of ammonification ($\text{NO}_x^- + \text{NH}_4^+$) and nitrification (NO_x^-) in the shorter incubations i.e., the 685. For the ambient ¹⁴N setting, the turnover of N₂O per day is clearly trivial (see Table 2 below).

Table 2. Estimating production of ¹⁴N₂O, mineralisation, nitrification and turnover of N₂O.

Mineralisation ($\mu\text{M N d}^{-1}$)	Nitrification ($\mu\text{M d}^{-1}$)	Yield of N ₂ O [†] (%)	N ₂ O ($\mu\text{M d}^{-1}$)	Ambient N ₂ O (μM)	Turnover of N ₂ O (d^{-1})
0.110	0.107	0.5	0.00053	0.012	0.044

[†]Our median overnight O₂ saturation was 80%, so here we use Goreau’s et al 1980 yield of 0.5% at 50% oxygen.

We can then extend this estimate using the ¹⁵N atom % of PON after incubating biomass with ¹⁵N₂ and scale mineralisation and nitrification to the fraction of ¹⁵N-PON (i.e., F_{PON}). This suggests that production of ¹⁵N₂O by recycling ¹⁵N would be in the fM d⁻¹ range and $\sim F_{\text{N}_2\text{O}}$ (i.e., maximum ¹⁵N-labelling after 1 day) would be 9.8×10^{-7} i.e., practically nothing. Including this would add little of any real value and would likely only confound understanding of Table 1.

Figure 3c: It would be instructive to include the production of N_2 in this plot.

Fig. 3c (now Fig. 3b) is from the additional incubations and the main purpose was to see whether N_2O can be presented as dissolved inorganic nitrogen, without the interference of formaldehyde. We did not measure the production of N_2 here. However, the set up for the additional incubations are the same as our other incubations and we have discussed the production of N_2 in Fig. 2c. So it does not feel needed to discuss this again here in Fig.3.

Figure 3d: I know this is a simplified diagram, but since you have N_2O reduction to N_2 via denitrification in many/half of your experiments, should this diagram also contain nitrate reduction to nitrite and N_2O via denitrification?

We do not think so as the purpose here is to simply illustrate direct and indirect N_2O fixation and the subsequent fate of that NH_4^+ . The added complexity through including NO_3^- and NO_2^- reduction would not add anything to that main message.

Supplementary figure 3: What kind of linear regression is the line in a?

Thanks for spotting this, the text has been revised to “The solid lines in all panels are simple first order linear regressions”. This is now Supplementary Fig. 2.

References

Kraft, B., N. Jehmlich, M. Larsen, L. A. Bristow, M. Könneke, B. Thamdrup, and D. E. Canfield. 2022. Oxygen and nitrogen production by an ammonia-oxidizing archaeon. *Science*. doi:10.1126/science.abe6733

Redfield, A. C. 1934. On the Proportions of Organic Derivatives in Sea Water and Their Relation to the Composition of Plankton, University Press of Liverpool.

Smith, C., Z. R. J. Nicholls, K. Armour, and others. 2021. The Earth’s Energy Budget, Climate Feedbacks, and Climate Sensitivity Supplementary Material, In V. Masson-Delmotte, P. Zhai, A. Pirani, et al. [eds.], *Climate Change 2021: The Physical Science Basis. Contribution of Working Group I to the Sixth Assessment Report of the Intergovernmental Panel on Climate Change*. Cambridge University Press.

Sun, X., A. Jayakumar, J. C. Tracey, E. Wallace, C. L. Kelly, K. L. Casciotti, and B. B. Ward. 2021. Microbial N_2O consumption in and above marine N_2O production hotspots. *ISME J.* 15: 1434–1444. doi:10.1038/s41396-020-00861-2

Reviewer #2 (Remarks to the Author):

I find the research topic of this manuscript very significant and provides really interesting results on the capacity of N_2O fixation by freshwater communities in N-limited experimental ponds. I also find very interesting the comparison of N_2O fixation with canonical N_2 fixation and their dependencies on seasonality and temperature, as well as the search for potential candidates in the diazotrophic community as N_2O sinks. In addition, the development of the experimental designs was very well elaborated and thought out.

My main concern is that although in terms of structure the article is well written, it is tremendously long and many parts of the text in the results and methods sections can be shortened. Articles should be written in a concise manner showing only the main results. Therefore, I strongly suggest reducing both sections without compromising critical information and shuttle the redundant or unnecessary information to the supplementary section. For example, there are sentences in the results that can be eliminated because they are over-understood by looking at the figures or because they are repetitive with the information in the figure captions (in fact, I suggest reducing the figure captions as well). Also, the methodology should not take up more than 5-6 pages of the main document and here there are 16 pages!

Point taken - the PhD (Yueyue) was just being diligent with their first paper and especially given the novelty of what we are reporting. Following the Nature Communication author guidelines, we have edited our Methods down from ~4,800 to the expected ~3,000 and shortened other parts of the text.

On the other hand, it is not clear from the main text when the samples were taken from the ponds for each of the analyses. I deduce from some figures that the moments and/or sampling times were not always the same for each analysis, right? This information must be explicit in the methodology for each analysis. In some parts, the information on sampling times even seems contradictory or confusing. For example, the graphs b and c in Fig. 1 show a sampling period from January to December (what year?), but the title of the figure indicates that the sampling was from November 2019 to April 2022; therefore, it does not match. The same happens in Fig. S2 where the graph shows a sampling period for 12 months in a year, but the title of the figure indicates that the sampling was from January 2019 to December 2021 (this means 3 years of sampling!). Please, review carefully this information in all figures. I also suggest including a short paragraph at the beginning of the results contextualizing the experimental design and the sampling to understand the results.

As above for reviewer 1, we apologies for any confusion caused here as we were simply trying to show the data at representative, seasonal times of the year. The ponds are in Dorset which is a 2-3h train journey from Queen Mary in London and field work was seriously impacted by the pandemic and lock-down in March 2020. See the table below, sampling month is therefore not continuous and not in order of time, but scattered across Nov 2019 to Apr 2022 and has been rearranged by month – regardless of year – simply to plot the seasonal data. The overall effect of seasonal variation in temperature on either gas is still apparent in **d** and **e**, regardless of year. We have updated the legend for Fig. 1 in the manuscript to reflect this.

Table 1. Sampling dates (year & month) for gas saturations.

Year	Month	Gas measured
2019	11	N ₂ O
2020	8, 9, 12	N ₂ O, N ₂
2021	6, 7, 9, 10	N ₂ O, N ₂
2022	1, 2, 3, 4	N ₂ O, N ₂

Finally, the discussion seems to me well raised and bounded to the results obtained. Two suggestions:

L388: Delete "in most" (only leave the percentage)

Done

L433-436: N fixation (even high rates) has been demonstrated in N-enriched aquatic systems, so this sentence is not correct. Please, rewrite the sentence.

This was an over-simplification, and the paragraph has now been completely revised. See new lines 427 – 439.

Other notes

The x-axis in Fig.2a had been mislabelled and the order of the months confused. All three panels comprising Fig. 2 have now been revised and the source data updated.

We have also changed the text “N₂O consumption” in Fig. 5b to “N₂O reduction” to be consistent throughout the manuscript.

References

- Barneche, D.R., Hulatt, C.J., Dossena, M., Padfield, D., Woodward, G., Trimmer, M. and Yvon-Durocher, G. 2021. Warming impairs trophic transfer efficiency in a long-term field experiment. *Nature* 592(7852), 76-79.
- Broda, E. 1977. Two kinds of lithotrophs missing in nature. *Zeitschrift für allgemeine Mikrobiologie* 17(6), 491-493.
- Butler, J.H., Elkins, J.W., Thompson, T.M. and Egan, K.B. 1989. Tropospheric and dissolved N₂O of the west Pacific and east Indian Oceans during the El Niño Southern Oscillation event of 1987. *Journal of Geophysical Research: Atmospheres* 94(D12), 14865-14877.
- Cline, J.D., Wisegarver, D.P. and Kelly-Hansen, K. 1987. Nitrous oxide and vertical mixing in the equatorial Pacific during the 1982–1983 El Niño. *Deep Sea Research Part A. Oceanographic Research Papers* 34(5-6), 857-873.
- Falkowski, P. 1983 *Enzymology of Nitrogen Assimilation Nitrogen in the Marine Environment*, Academic Press.
- Henry, S., Bru, D., Stres, B., Hallet, S. and Philippot, L. 2006. Quantitative detection of the *nosZ* gene, encoding nitrous oxide reductase, and comparison of the abundances of 16S rRNA, *narG*, *nirK*, and *nosZ* genes in soils. *Appl. Environ. Microbiol.* 72(8), 5181-5189.
- Knapp, A. 2012. The sensitivity of marine N₂ fixation to dissolved inorganic nitrogen. *Frontiers in microbiology* 3, 374.
- Kuypers, M.M., Marchant, H.K. and Kartal, B. 2018. The microbial nitrogen-cycling network. *Nature Reviews Microbiology* 16(5), 263.
- Mulder, A., Van de Graaf, A.A., Robertson, L. and Kuenen, J. 1995. Anaerobic ammonium oxidation discovered in a denitrifying fluidized bed reactor. *FEMS microbiology ecology* 16(3), 177-183.
- Nguyen, N.-L., Yu, W.-J., Yang, H.-Y., Kim, J.-G., Jung, M.-Y., Park, S.-J., Roh, S.-W. and Rhee, S.-K. 2017. A novel methanotroph in the genus *Methylomonas* that contains a distinct clade of soluble methane monooxygenase. *Journal of Microbiology* 55, 775-782.
- Nicholls, J.C., Davies, C.A. and Trimmer, M. 2007. High-resolution profiles and nitrogen isotope tracing reveal a dominant source of nitrous oxide and multiple pathways of

- nitrogen gas formation in the central Arabian Sea. *Limnology and oceanography* 52(1), 156-168.
- Rees, A.P., Brown, I.J., Jayakumar, A., Lessin, G., Somerfield, P.J. and Ward, B.B. 2021. Biological nitrous oxide consumption in oxygenated waters of the high latitude Atlantic Ocean. *Communications Earth & Environment* 2(1), 1-8.
- Rolando, J.L., Kolton, M., Song, T., Liu, Y., Pinamang, P., Conrad, R.E., Morris, J.T., Konstantinidis, K.T. and Kostka, J.E. 2023. Sulfur oxidation and reduction are coupled to nitrogen fixation in the roots of a salt marsh foundation plant species. *bioRxiv*, 2023.2005.2001.538948.
- Strous, M., Fuerst, J.A., Kramer, E.H., Logemann, S., Muyzer, G., van de Pas-Schoonen, K.T., Webb, R., Kuenen, J.G. and Jetten, M.S. 1999. Missing lithotroph identified as new planctomycete. *Nature* 400(6743), 446-449.
- Sun, X., Jayakumar, A., Tracey, J.C., Wallace, E., Kelly, C.L., Casciotti, K.L. and Ward, B.B. 2021. Microbial N₂O consumption in and above marine N₂O production hotspots. *The ISME Journal* 15(5), 1434-1444.
- Thamdrup, B. and Dalsgaard, T. 2000. The fate of ammonium in anoxic manganese oxide-rich marine sediment. *Geochimica et Cosmochimica Acta* 64(24), 4157-4164.
- Thamdrup, B. and Dalsgaard, T. 2002. Production of N₂ through anaerobic ammonium oxidation coupled to nitrate reduction in marine sediments. *Applied and environmental microbiology* 68(3), 1312-1318.
- Trimmer, M., Nicholls, J.C. and Deflandre, B. 2003. Anaerobic ammonium oxidation measured in sediments along the Thames estuary, United Kingdom. *Appl Environ Microbiol* 69(11), 6447-6454.
- Trimmer, M., Shelley, F.C., Purdy, K.J., Maanoja, S.T., Chronopoulou, P.-M. and Grey, J. 2015. Riverbed methanotrophy sustained by high carbon conversion efficiency. *The ISME journal* 9(10), 2304-2314.
- Weiss, R. and Price, B. 1980. Nitrous oxide solubility in water and seawater. *Marine chemistry* 8(4), 347-359.
- Zhu, Y., Purdy, K.J., Eyice, Ö., Shen, L., Harpenslager, S.F., Yvon-Durocher, G., Dumbrell, A.J. and Trimmer, M. 2020. Disproportionate increase in freshwater methane emissions induced by experimental warming. *Nature Climate Change*, 1-6.

Reviewer #1 (Remarks to the Author):

General comments

In their revised text and response to reviewers, the authors addressed most of my concerns.

Upon revising their detection limit for $^{15}\text{N}_2$ production, the authors find that $^{15}\text{N}_2$ production was only present in 24% of their incubations, and only in those with benthic biomass. This result is less surprising than the authors' original finding that $^{15}\text{N}_2$ production was present in half of their incubations, and I retract my recommendation for the authors to emphasize this particular finding. This revised estimate also reduces my concern about indirect $^{15}\text{N}_2\text{O}$ fixation.

I also appreciate the authors' attempt to spell out their calculations in Table 1 a bit more. Did the maximum rate of $^{15}\text{N}_2$ production via denitrification (and thus the maximum potential rate of indirect N_2O fixation) change when the authors revisited their $^{15}\text{N}_2$ data? If so, row three of Table 1 should be revised.

It would be helpful if the following text from the response to reviewers file was added to the caption of Supplementary Figure 5, or even the main text:

"On average, the production of $^{30}\text{N}_2$ from $^{45}\text{N}_2\text{O}$ in the benthic incubations was $0.27\ \mu\text{M}$ (Fig. 3b, below), which would result in $0.29\ \text{nmol N g}^{-1}\ \text{d}^{-1}$ of ^{15}N assimilation if all of that $^{30}\text{N}_2$ was subsequently fixed. In contrast, this $0.29\ \text{nmol N}$ is far lower than the rate of ^{15}N assimilation that we measured ($5.3\ \text{nmol N g}^{-1}\ \text{d}^{-1}$) with $^{15}\text{N}_2\text{O}$, which means that even in the benthic biomass incubations that showed significant $^{15}\text{N}_2$ production, $^{15}\text{N}_2\text{O}$ assimilation still appears to have been mainly direct."

I appreciate the authors' efforts to describe the $^{45}\text{N}_2\text{O}$ and $^{46}\text{N}_2\text{O}$ measurements in more detail. For the scope of this paper, I acknowledge that the prevailing practice in the field does not involve regular calibration of the IRMS using a ^{15}N -labeled N_2O reference material. Nonetheless, consistent calibration of the IRMS using reference materials of varying $^{45}\text{N}_2\text{O}$ content, even in the context of tracer studies, warrants integration into future protocols.

Technical corrections

Line 22: should this be, "the regulation of N_2O -fixation compared to..."

Line 201: typo

Line 206: "and water column" meaning unclear

Line 310: "we sort to answer" meaning unclear

Line 404: change to "are probably associated"

Detailed replies (in blue) to reviewer comments (in black) on NCOMMS-23-18678A

Reviewer #1 (Remarks to the Author):

General comments

In their revised text and response to reviewers, the authors addressed most of my concerns.

Upon revising their detection limit for $^{15}\text{N}_2$ production, the authors find that $^{15}\text{N}_2$ production was only present in 24% of their incubations, and only in those with benthic biomass. This result is less surprising than the authors' original finding that $^{15}\text{N}_2$ production was present in half of their incubations, and I retract my recommendation for the authors to emphasize this particular finding. This revised estimate also reduces my concern about indirect $^{15}\text{N}_2\text{O}$ fixation.

Thank you for all your help in improving our manuscript. We are glad that it is clearer now.

I also appreciate the authors' attempt to spell out their calculations in Table 1 a bit more. Did the maximum rate of $^{15}\text{N}_2$ production via denitrification (and thus the maximum potential rate of indirect N_2O fixation) change when the authors revisited their $^{15}\text{N}_2$ data? If so, row three of Table 1 should be revised.

Thank you. The maximum rate of $^{15}\text{N}_2$ production i.e., maximum indirect N_2O fixation was calculated based on the rate of total $^{15}\text{N}_2\text{O}$ consumption and is not affected by the $^{15}\text{N}_2$ data. Therefore, there is no need to revise Table 1.

It would be helpful if the following text from the response to reviewers file was added to the caption of Supplementary Figure 5, or even the main text:

"On average, the production of $^{30}\text{N}_2$ from $^{45}\text{N}_2\text{O}$ in the benthic incubations was $0.27 \mu\text{M}$ (Fig. 3b, below), which would result in $0.29 \text{ nmol N g}^{-1} \text{ d}^{-1}$ of ^{15}N assimilation if all of that $^{30}\text{N}_2$ was subsequently fixed. In contrast, this 0.29 nmol N is far lower than the rate of ^{15}N assimilation that we measured ($5.3 \text{ nmol N g}^{-1} \text{ d}^{-1}$) with $^{15}\text{N}_2\text{O}$, which means that even in the benthic biomass incubations that showed significant $^{15}\text{N}_2$ production, $^{15}\text{N}_2\text{O}$ assimilation still appears to have been mainly direct."

Thanks for the suggestion. We have added the text to the legend of supplementary Figure 5. For the main text, however, we already present two clear lines of argument for direct N_2O fixation: $^{15}\text{N}_2$ production was not significant in the floating biomass (lines 172 – 176), yet ^{15}N fixation from $^{15}\text{N}_2\text{O}$ was; and the potential maximum rate for indirect $^{15}\text{N}_2\text{O}$ fixation was much lower than the measured rate of $^{15}\text{N}_2\text{O}$ fixation i.e., it must be direct (lines 182 – 191). This second point is reiterated in the legend for Table 1 which we now believe is as clear as it can be. The text "quoted" [note the use $^{45}\text{N}_2\text{O}$ and not our original $^{15}\text{N}_2\text{O}$] above is related as it makes a similar argument against indirect N_2O fixation in relation to $^{15}\text{N}_2$ production from $^{15}\text{N}_2\text{O}$. This is, however, distinct from the argument in the main text and Table 1 which is based solely on the measured rates of $^{15}\text{N}_2\text{O}$ reduction i.e., we know the rate of $^{15}\text{N}_2\text{O}$ reduction, so how much $^{15}\text{N}_2$ is that equivalent to and how much ^{15}N -fixation could that sustain? Presenting both in the main text would require a lot of extraneous explanation that would not add anything to the argument presented in the main text.

I appreciate the authors' efforts to describe the $^{45}\text{N}_2\text{O}$ and $^{46}\text{N}_2\text{O}$ measurements in more detail. For the scope of this paper, I acknowledge that the prevailing practice in the field does not involve regular calibration of the IRMS using a ^{15}N -labeled N_2O reference material. Nonetheless, consistent calibration of the IRMS using reference materials of varying $^{45}\text{N}_2\text{O}$ content, even in the context of tracer studies, warrants integration into future protocols.

We agree. Regular calibration of IRMS with known purities of ^{15}N - N_2O would be good.

Technical corrections

Line 22: should this be, “the regulation of N_2O -fixation compared to...”
Edited, see new line 17.

Line 201: typo
Corrected. See “measurments” to “measurements” in new line 199.

Line 206: “and water column” meaning unclear
We have changed this to “or free-living in the water column”. See new lines 204 – 205.

Line 310: “we sort to answer” meaning unclear
We have edited this sentence. See new lines 309 – 310.

Line 404: change to “are probably associated”
Done.